# Structural basis of R-loop recognition by the S9.6 monoclonal antibody

Charles Bou-Nader [1], Ankur Bothra[2], David N. Garboczi[3], Stephen H. Leppla[2]✉ & Jinwei Zhang [1]✉

R-loops are ubiquitous, dynamic nucleic-acid structures that play fundamental roles in DNA replication and repair, chromatin and transcription regulation, as well as telomere maintenance. The DNA-RNA hybrid–specific S9.6 monoclonal antibody is widely used to map R-loops. Here, we report crystal structures of a S9.6 antigen-binding fragment (Fab) free and bound to a 13-bp hybrid duplex. We demonstrate that S9.6 exhibits robust selectivity in binding hybrids over double-stranded (ds) RNA and in categorically rejecting dsDNA. S9.6 asymmetrically recognizes a compact epitope of two consecutive RNA nucleotides via their 2'-hydroxyl groups and six consecutive DNA nucleotides via their backbone phosphate and deoxyribose groups. Recognition is mediated principally by aromatic and basic residues of the S9.6 heavy chain, which closely track the curvature of the hybrid minor groove. These findings reveal the molecular basis for S9.6 recognition of R-loops, detail its binding specificity, identify a new hybrid-recognition strategy, and provide a framework for S9.6 protein engineering.

[1] Laboratory of Molecular Biology, National Institute of Diabetes and Digestive and Kidney Diseases, Bethesda, MD 20892, USA. [2] Laboratory of Parasitic Diseases, National Institute of Allergy and Infectious Diseases, Bethesda, MD 20892, USA. [3] Structural Biology Section, Research Technologies Branch, National Institute of Allergy and Infectious Diseases, Bethesda, MD 20892, USA. ✉email: sleppla@niaid.nih.gov; jinwei.zhang@nih.gov

R-loops are three-stranded nucleic acid structures consisting of a DNA-RNA hybrid duplex and a single-stranded (ss) DNA strand locally displaced by the RNA strand[1]. R-loops are prevalent in bacterial, eukaryotic, and viral genomes, and cover ~ 5, 8, and 10% of the human, yeast, and *Arabidopsis* genomes, respectively[2–4]. Extensive research in recent years has shed light on the formation, biological function, and regulation of R-loops, and has provided new tools for their detection.

R-loops can form both co-transcriptionally and post-transcriptionally. During transcript elongation, multi-subunit RNA polymerases (RNAPs) clamp a 9–10-bp (base pair)-long DNA-RNA hybrid while displacing and extruding the non-template strand DNA, sustaining a transcription bubble traveling downstream along double-stranded (ds) DNA[5,6]. Emerging from the RNA-exit channel of RNAP, nascent RNA transcripts remain in proximity to and can reanneal with the template DNA strand forming R-loops in *cis*. These co-transcriptionally formed R-loops range from ~60 to 2000 bp in length[7,8]. Away from the transcription complexes, ssRNA strands can also invade complementary dsDNA or anneal with exposed ssDNA strands. Notable examples of R-loop formation in *trans* include guide RNA annealing to the target DNA strand in CRISPR-Cas9 to trigger double-stranded breaks[9], RAD51-dependent, telomeric repeat-containing RNA (TERRA) long noncoding RNA (lncRNA) recruitment to short telomeres needing maintainence[10], and GAL lncRNAs hybridizing across *GAL* DNA to induce gene looping and transcription derepression during metabolic adaptation[11].

R-loops are dynamic, reversible structures that can play both positive and negative roles that impact cellular metabolism, and therefore are subjected to stringent control. R-loops are enriched near both ends of transcriptional units including promoter and terminator regions, and on highly expressed genes[4,12], presumably due to a more open chromatin structure. Unscheduled, harmful R-loops emerge as by-products of transcription and can cause genome instability and blockage of replication[13] and transcription[14,15]. They are therefore either cleaved by RNases H1[16] or H2[17] or resolved by DNA topoisomerases[18] or helicases such as DHX9[19] and senataxin[20,21]. Recently, R-loops have also been recognized as functional structures that exert programmed, regulatory roles in chromatin structure, telomere maintenance, DNA replication and repair, as well as transcription initiation, elongation, and termination, etc[7,22,23]. Intriguingly, R-loops provide a primary route of entry for lncRNAs to localize to and act on specific DNA loci[10,24]. Consistent with their critical importance to normal cellular metabolism, R-loop dysregulation has been linked to severe human diseases including amyotrophic lateral sclerosis type 4 (ALS4)[25], ataxia oculomotor apraxia type 2 (AOA2)[25], Aicardi-Goutière Syndrome (AGS)[26], and others[23,25].

Given the importance of R-loop function and regulation in biology and disease, several methods have been developed to detect and characterize them genome wide[27]. Most methods, such as DRIPc-seq, rely on immunodetection of R-loops by the S9.6 monoclonal antibody which specifically binds DNA-RNA hybrids[28–30]. However, despite a decade of widespread use, it remains unknown how selective S9.6 is for hybrids over dsRNA and dsDNA, whether S9.6 possesses intrinsic sequence specificity, and what is the molecular basis underlying its selectivity. Indeed, significant concerns have been raised about its cross-reactivity with the more abundant dsRNA, particularly in imaging applications[31]. Some sequence bias was also reported[32]. Recently, S9.6-independent R-loop detection methods have been developed, which employ either a non-cleaving RNase H domain to recognize the hybrids (e.g., DRIVE-seq and R-ChIP)[33–35] or bisulfite treatment or activation-induced cytidine deaminase (AID) (e.g., SMRF-seq[36]) to chemically or enzymatically modify and mark the

ssDNA strands of R-loops[37]. However, these alternative methods have their own shortcomings such as inefficient R-loop detection and cannot wholly replace S9.6-based methods[27,30,35,38]. Therefore, it is important to understand the recognition strategy and binding specificities of S9.6. To this end, we determined the crystal structures of an antigen-binding fragment (Fab) of S9.6 and its complex with a 13-bp DNA-RNA hybrid. We further quantitatively assessed the binding preferences of S9.6 for various duplex structures and sequences and characterized the S9.6-hybrid interface. Our findings show that S9.6 possesses strong intrinsic specificity for hybrids, asymmetrically recognizes the RNA and DNA strands, and exhibits preferences for binding GC-rich sequences.

## Results

**S9.6 exhibits specificity for DNA-RNA hybrid duplex over dsDNA and dsRNA.** To assess the binding preferences of the S9.6 antibody for different types of double-stranded nucleic acids, we monitored the interaction of an S9.6 Fab with a 13-bp dsRNA, dsDNA, or DNA-RNA hybrid duplex of the same length and sequence using three biophysical techniques (Fig. 1).

First, we verified that these 13-bp nucleic acid duplexes are stable at room temperature (~21 °C) using circular dichroism (CD). CD spectra of all three assemblies exhibited signature bands of duplex nucleic acids at ~209, 262, and 280 nm (Supplementary Fig. 1), and showed that the hybrid structure possesses characteristics of both dsDNA and dsRNA and is closer to dsRNA, as reported previously[39]. Further, we measured the thermostability of these duplexes by differential scanning calorimetry (DSC), which produced $T_m$ values of 58, 55, and 72 °C for dsDNA, hybrid, and dsRNA, respectively. Thus, both CD and DSC analyses confirm that the three 13-bp duplex nucleic acids of the particular sequence used are stable at 21 °C.

Size-exclusion chromatography coupled to multi-angle light scattering (SEC-MALS) revealed a distinctive peak representing a stable 1:1 complex formed between the hybrid and S9.6 (Fig. 1c, green). By contrast, no complex peak was observed when S9.6 was mixed with either dsRNA or dsDNA, suggesting a strong preference of S9.6 for the hybrid. This shows that only hybrid duplexes can form complexes with S9.6 that are stable enough to survive gel filtration. To quantify S9.6 binding affinities for the three classes of nucleic acid duplexes, we labeled the 13-bp nucleic acid substrates with FAM (fluorescein) on their 5′ ends and monitored the increase of fluorescence polarization (FP) induced by S9.6 binding. When S9.6 bound to the labeled DNA-RNA hybrid, robust changes in fluorescence polarization were observed that indicated an apparent $K_d$ of 232 nM (Fig. 1d and Supplementary Table 1). The binding of S9.6 to labeled dsRNA yielded a 16 times greater apparent $K_d$ of 3800 nM, and the binding of S9.6 to dsDNA was not detectable in this assay. Lastly, preferred hybrid binding by S9.6 was confirmed by isothermal titration calorimetry (ITC), which yielded a $K_d$ of ~415 nM for the hybrid, but only insignificant heat with dsRNA titration (Fig. 1f, g). ITC also affirms 1:1 stoichiometric binding and reveals that binding is enthalpically driven. The $K_d$ values are generally higher compared to the reported affinities of S9.6 for hybrids, estimated as 0.6–1.8 nM by surface plasmon resonance (SPR)[30] and 3.1–100 nM by microscale thermophoresis (MST) and gel shift[32]. Variations in methodologies, sources, and forms of S9.6, hybrid sequences, lengths, and structures, as well as buffer compositions and surface and immobilization effects likely account for the range of measured $K_d$ values. IgG forms of S9.6 bearing two Fab domains are expected to bind more avidly than single Fabs or single-chain variable fragment (scFv). Indeed, we observed similar affinities between Fab and scFv forms of S9.6

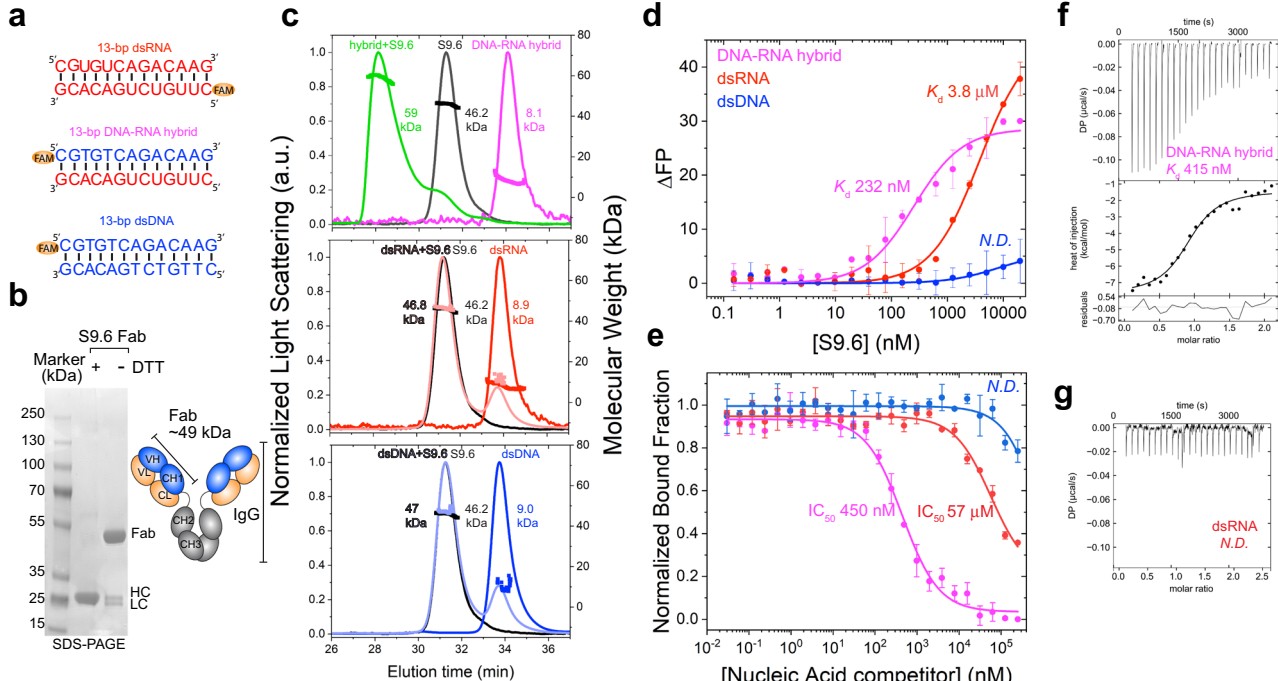

**Fig. 1 Biophysical characterizations of nucleic acid binding preferences of S9.6 Fab. a** Sequences of double-stranded (ds) nucleic acids used for S9.6 characterizations and locations of FAM labels. **b** Schematic representation and denaturing SDS-PAGE analysis of S9.6 Fab in oxidizing and reducing conditions. SDS-PAGE analysis was performed twice. HC heavy chain, LC light chain. **c** SEC-MALS profiles of free S9.6 Fab (black), free DNA-RNA hybrid (magenta), free dsRNA (red), and free dsDNA (blue) and stoichiometric (1:1) mixtures of S9.6 Fab with hybrids (green), dsRNA (salmon), and dsDNA (light blue). SEC-MALS-derived molecular weights are indicated. **d** Binding affinity measurements of S9.6 Fab with nucleic acids in **a** by fluorescence polarization titration. Apparent binding constants ($K_d$s) are indicated. Values are mean ± s.d. $n = 3$ biologically independent samples. ΔFP: changes in fluorescence polarization, in mP units. ND: not determined. **e** Competition experiments of fluorescently labeled DNA-RNA hybrids bound to S9.6 with increasing amounts of unlabeled nucleic acids, colored as in **d**. Apparent $IC_{50}$s are indicated. Values are mean ± s.d. $n = 3$ biologically independent samples. **f** Isothermal titration calorimetry (ITC) profile of S9.6 binding to DNA-RNA hybrid. The measured binding constant is indicated. Values are mean ± s.d. $n = 2$ biologically independent samples. **g** ITC profile of S9.6 binding to dsRNA. Source data are provided as a Source Data file.

(Supplementary Fig. 3). Nevertheless, these measurements reveal a 16-fold lower $K_d$ for S9.6 binding to hybrids and thus, much tighter binding to a hybrid than to dsRNA, in congruence with the SEC-MALS analysis. To ask if this binding preference of S9.6 is conserved towards different hybrid sequences, we measured S9.6 binding to two naturally occurring R-loop sequences at the *FUS* locus and β-actin terminator[36,40]. In both cases, similar S9.6 preferences were observed (Supplementary Fig. 2), suggesting that S9.6 has a general propensity to preferably bind hybrids over dsRNA and has little affinity for dsDNA.

Finally, to examine the preferences of S9.6 for various duplexes in an environment where the hybrids coexist with other nucleic acids, we measured the competition between the FAM-labeled hybrids pre-bound to S9.6 and increasing amounts of unlabeled duplexes (Fig. 1e). The unlabeled hybrid effectively competed for S9.6 binding with an apparent half-maximal inhibitory concentration ($IC_{50}$) of 450 nM comparable to the observed $K_d$. dsRNA was a much weaker competitor with a ~127-fold higher $IC_{50}$ of 57 μM, while dsDNA was unable to outcompete the hybrid even in 50,000-fold excess. Collectively, these data reveal an intrinsic preference for S9.6 to bind hybrid duplexes over dsRNA (16–127-fold) and no significant affinities towards dsDNA. This specificity is on par with estimates for RNases H[27,41].

**Overall structures of free and hybrid-bound S9.6 Fab.** To understand the specificity of S9.6 for hybrids over dsRNA and dsDNA, we solved crystal structures of the free S9.6 Fab at 2.3 Å and its complex with a 13-bp hybrid at 3.1 Å resolution (Fig. 2,

Supplementary Fig. 4, and Supplementary Table 2). In both structures, the heavy and light chains of the Fab stabilize each other through extensive hydrophobic interactions primarily between aromatic side chains (Tyr, Phe, Trp, etc.), burying a total solvent-accessible surface area of 1783 Å² (Supplementary Fig. 5). The heavy and light chains of the Fab are covalently linked by a disulfide bond between C219$^L$ ($^L$ indicates residues from the light chain) near the C-terminus of the CL and C134 on CH1, as evidenced by the disruption of the interchain linkage by a C219A substitution or dithiothreitol (DTT) treatment (Fig. 1b and Supplementary Fig. 3). This interchain disulfide bond is mobile as it lacks well-defined electron density and does not contribute to hybrid binding (Supplementary Fig. 3h). This is consistent with the robust network of aromatic and hydrophobic contacts that maintain the Fab fold in the absence of the interchain disulfide linkage.

The co-crystal structure of the S9.6 Fab-hybrid complex reveals that S9.6 binding buries solvent-accessible surface areas of 450 Å² with the RNA strand and 382 Å² with the DNA strand, mostly through interactions with the VH domain (Fig. 2c and Supplementary Fig. 5). By contrast, the VL makes few direct contacts to the hybrid and primarily acts to constrain and help present the complementarity-determining regions (CDRs) of the VH. Hybrid binding does not induce significant conformational changes in the CDRs, suggesting pre-configuration of the paratope (Supplementary Fig. 6). The root-mean-square deviation (RMSD) between the two overall Fab structures is 1.8 Å over 379 Cα and between the variable regions is 0.49 Å over 210 Cα (Supplementary Fig. 6). While CDR-H1 (denoting the first CDR

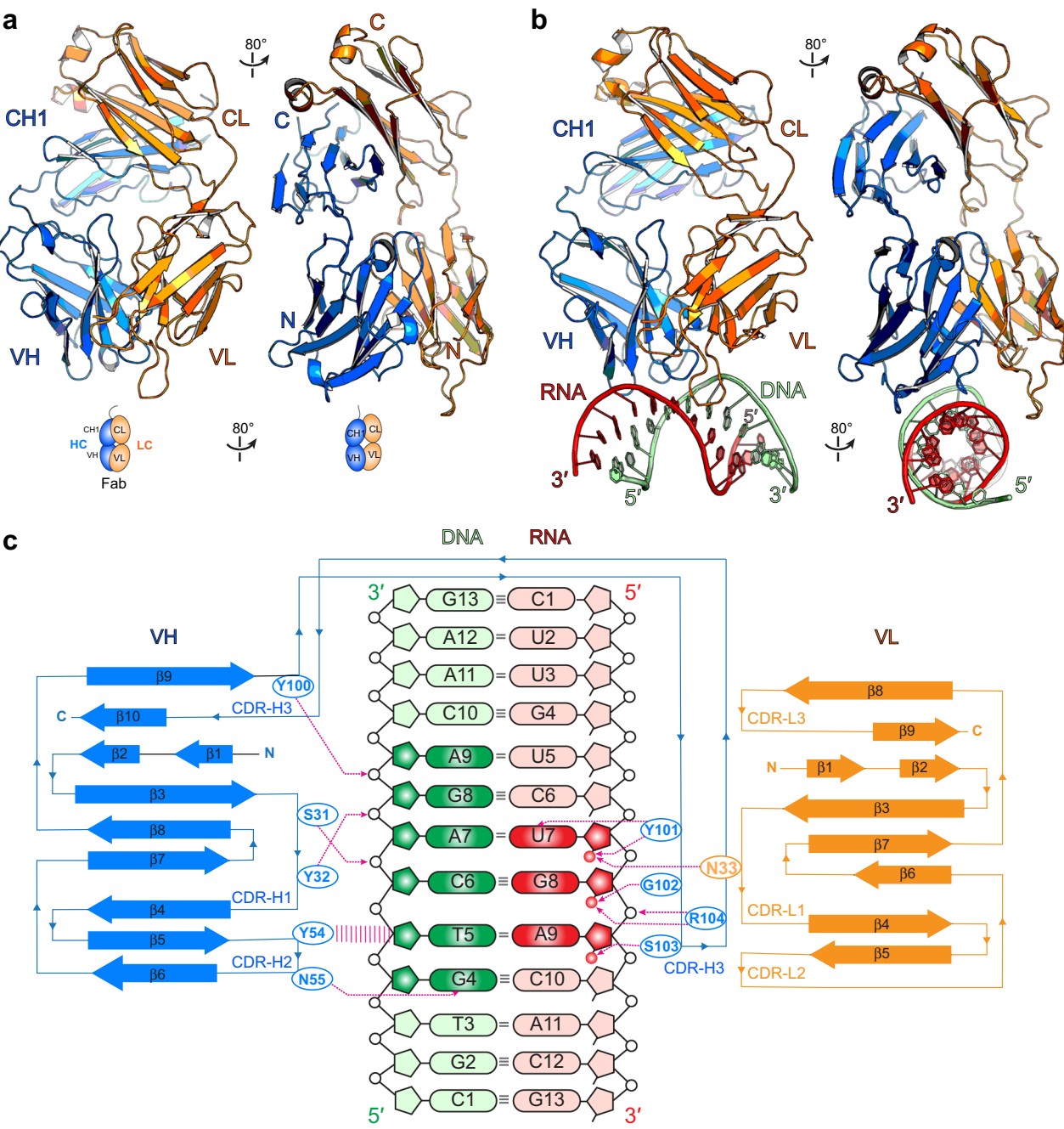

**Fig. 2 Overall structures of free S9.6 Fab and its complex with a 13-bp DNA-RNA hybrid. a** Structure and cartoon schematic of free S9.6 Fab. Heavy chain (HC) is in blue; light chain (LC) is in orange. VH and CH1: variable and first constant domains of the heavy chain; VL and CL: variable and constant domains of the light chain. N and C termini are indicated. **b** Co-crystal structure of S9.6 Fab bound to a 13-bp hybrid of RNA (red) and DNA (green) strands. **c** Diagram of secondary structure and interactions of S9.6 with the hybrid. Nucleotides in direct contact are in a darker color. Hydrogen bonds are shown as dashed magenta lines; sugar–π packing interaction as parallel magenta lines. RNA 2′-hydroxyls in contact with S9.6 are shown as red spheres. Complementarity-determining regions (CDRs) H1–H3 and L1–L3 are denoted. The extended CDR-H3 loop hosts Y101-R104 residues for RNA strand recognition.

of VH), H2, L1, L2, and L3 are nearly identical between both structures, the loop of CDR-H3 is more extended in the absence of the hybrid, with Y100 and Y101 in the loop shifting by ~4 Å upon hybrid binding (Supplementary Fig. 6). This more relaxed CDR-H3 loop conformation appears to represent the resting, low-energy state in solution, as two different crystal forms with distinct crystal packing contacts exhibited the same conformation. Conformational rigidification of the CDRs often correlates with increased binding during affinity maturation of antibodies,

due to reduced entropic cost associated with binding[42]. This observed pre-configuration of the S9.6 CDRs is likely at least partially responsible for the high-affinity binding of S9.6 to hybrids.

**S9.6 recognizes tandem 2′-hydroxyl groups on the RNA strand.**
The co-crystal structure reveals that the S9.6 binding interface on the hybrid spans only the central 6-bp duplex out of the 13-bp used for crystallization (Fig. 2c), consistent with a previous report

that the minimal epitope required for S9.6 recognition is a 6-bp hybrid[30]. The 6-bp hybrid duplex is asymmetrically recognized, with three consecutive RNA nucleotides and six successive DNA nucleotides seen in direct contact with S9.6.

The RNA strand is recognized principally by CDR-H3, which is assisted by CDR-L1 and CDR-L3 (Figs. 2c, 3a, b and Supplementary Fig. 5). The ten-residue loop of CDR-H3 extends into the minor groove of the hybrid, which positions a stretch of four consecutive residues—Y101, G102, S103, and R104—to recognize three successive 2′-hydroxyls of the riboses of rU7, rG8, and rA9 (RNA strand numbering preceded by an "r"). This extended conformation of the CDR-H3 loop is apparently stabilized by interactions with the adjacent 14-residue loop of CDR-L1 (Fig. 3a, b). Starting from the N-terminal portion of the CDR-H3 loop, Y101 is inserted deep inside the minor groove (Supplementary Fig. 5), with its aromatic ring nearly co-planar with the rU7-dA7 base pair. From this location, Y101 makes three types of potential interactions with the hybrid: (a) it uses its hydroxyl group to hydrogen bond with the 2′-OH and the O2 atom of rU7 (Fig. 3a, b); (b) across the base pair from rU7, the edge of the Y101 benzene ring makes van der Waals interactions with the nucleobase of dA7 (distance: 3.8 Å); and (c) Y101 is also well-positioned to engage a possible stacking interaction with rG8 in a parallel-displaced configuration (Fig. 3a), enabled by the local under-twisting of the duplex from 32° to 20°. To understand which of the three types of Y101 contacts are important for hybrid binding, we generated Y101A and Y101F mutant Fabs. While the Y101A substitution abrogated hybrid binding entirely, Y101F had less than a twofold diminished binding (Fig. 3c, d). This suggests that the hydrogen bonds involving the Y101 hydroxyl contribute little to binding, including the contact to the O2 of rU7 as one of the two nucleobase-selective interactions. Consistent with this, when we swapped the dA7-rU7 pair with dT7-rA7, thus removing the O2 group, no binding defect was observed (Supplementary Fig. 7). Therefore, either the hydrophobic interaction with dA7, stacking interaction with rG8, or both, is important for binding.

Next to Y101 is G102 located at the tip of the β-hairpin loop forming a reverse turn[43]. The main-chain carbonyl of G102 contacts the 2′-OH of rG8, in addition to providing the backbone torsion angles enabling the reverse turn (Fig. 3a, b). G102A substitution reduced hybrid binding by 27-fold while G102L completely abolished it (Fig. 3c, d). This is likely due to the destruction of the reverse turn in both variants, but could also be partially driven by the loss of the rG8 contact. The more dramatic impact of G102L is probably due to steric conflict of the bulkier side chain with the rG8 ribose. The side chain of S103 contacts the 2′-OH of rA9 but contributes little to binding, since neither S103A nor S103L substitution produced a defect (Fig. 3d). R104 is another essential residue required for hybrid binding. It uses its guanidinium group to make a bivalent interaction with the 2′-OH of rG8 and the bridging phosphate oxygen between rG8 and rA9. Consequently, an R104A substitution eliminated binding (Fig. 3c, d).

Since the S103–rA9 2′-OH contact seems dispensable, we further examined the ribose requirements on the RNA strand for S9.6 recognition. We first asked if two consecutive 2′-OHs would suffice in driving S9.6 binding instead of three observed 2′-OH contacts. Indeed, such a chimeric hybrid harboring several tandem 2′-OHs bound S9.6 with a robust $K_d$ of 380 nM—only ~50% higher than the regular hybrid (Fig. 3e, f). Then, we asked if tandem 2′-OHs are required for S9.6 binding. Interestingly, a chimeric hybrid that contains alternating but no consecutive ribose and deoxyribose nucleotides on the RNA strand showed no significant binding with S9.6 ($K_d > 21$ μM, Fig. 3e, f), suggesting a requirement for tandem 2′-OHs. Finally, we asked if a single

stretch of three 2′-OHs would enable binding, and found such a construct only weakly bound S9.6 ($K_d > 7$ μM, Fig. 3e). Since this duplex is mostly DNA, we reasoned that it is likely forming a B-form helix incompatible with S9.6 binding, and confirmed this by its dsDNA-like CD spectra (Supplementary Fig. 1). These findings suggest that tandem 2′-OHs in the structural context of an A-form duplex are both necessary and sufficient for S9.6 recognition.

In contrast to the essential roles of the heavy chain CDRs, the light chain CDR-L1 and L3 serve mostly ancillary roles by constraining and presenting the principal paratopes on the heavy chain, whereas CDR-L2 does not contribute. H31[L] and Y37[L] of CDR-L1, and to a lesser extent Y101[L] of CDR-L3, form an aromatic enclosure to position R104 via cation–π interactions (Fig. 3a, b), whereas the main-chain carbonyls of G96[L] and S97[L] pin R104 down towards the RNA via two hydrogen bonds, completing the cage. On the opposite side of R104, H31[L] and Y37[L] also sandwich and position their intervening N33[L] to recognize the backbone of rU7 (Fig. 3a, b). Congruent with these observed contacts, H31A[L] and N33A[L] substitutions reduced hybrid binding by ten- and six-fold, respectively (Fig. 3c, d). Unlike N33A[L], the N33K[L] variant is fully functional, consistent with the electrostatic nature of the N33[L] contact, while a Y37F[L] substitution had only a twofold defect, consistent with Y37[L]'s cation–π interaction with R104.

Together, the structure and mutational analyses reveal that S9.6 recognizes the RNA strand through specific hydrogen bonds to tandem 2′-hydroxyls absent from the DNA strand. The interaction is primarily driven by CDR-H3, especially Y101 and R104, and the light chain employs several aromatic residues to help position these key residues for RNA recognition.

**S9.6 recognizes six backbone phosphates of the DNA strand.** Unlike the centralized interface with just two RNA nucleotides, S9.6 recognizes a stretch of six consecutive nucleotides of the DNA strand, primarily using heavy chain aromatic side chains especially Tyr (tyrosine) (Fig. 4a). Specifically, S31 and Y32 from CDR-H1 and Y100 from CDR-H3 hydrogen bond with the phosphate backbones of dA7, dG8, and dA9, respectively (Fig. 4a, b). The S31 contact itself is not important, since neither S31A nor S31G substitution led to any binding defects. However, an S31L mutation abolished S9.6 binding, likely due to a steric clash with the adjacent DNA phosphate backbone. The Y32A substitution weakened binding by ~100-fold, revealing its importance. By contrast, neither Y100 nor Y54[L] at the downstream edge of the epitope contributes to binding, as their substitutions caused no defect (Fig. 4d).

Moving upstream from the dA7-dG8-dA9 trinucleotide, the dT5 deoxyribose is recognized by Y54 of CDR-H2 via a sugar–π packing interaction (Fig. 4a, b). The critical importance of this contact is accentuated by a 14 and 58-fold reduction of binding affinity by the Y54A and Y54G substitutions, respectively (Fig. 4f). In congruence with the chemical nature of the sugar–π interaction[44], other aromatic residues or histidine also support binding, producing only minor defects by Y54F, Y54H, and Y54W substitutions (Fig. 4e, f). The slight preference for Tyr is in line with reports that Tyr is the most frequently used aromatic residue in sugar–π interactions for nucleic acid recognition by proteins, while Phe is preferred for π–π stacking[44]. All but one non-aromatic and non-histidine side chain substitutions at Y54 substantially reduced hybrid binding, by 4–58-fold. Interestingly, the Y54R mutation slightly enhanced binding, presumably by acquiring additional polar contacts with the negatively charged DNA phosphate backbone (Fig. 4f). Consistent with this charge complementarity, the neutral polar Y54Q

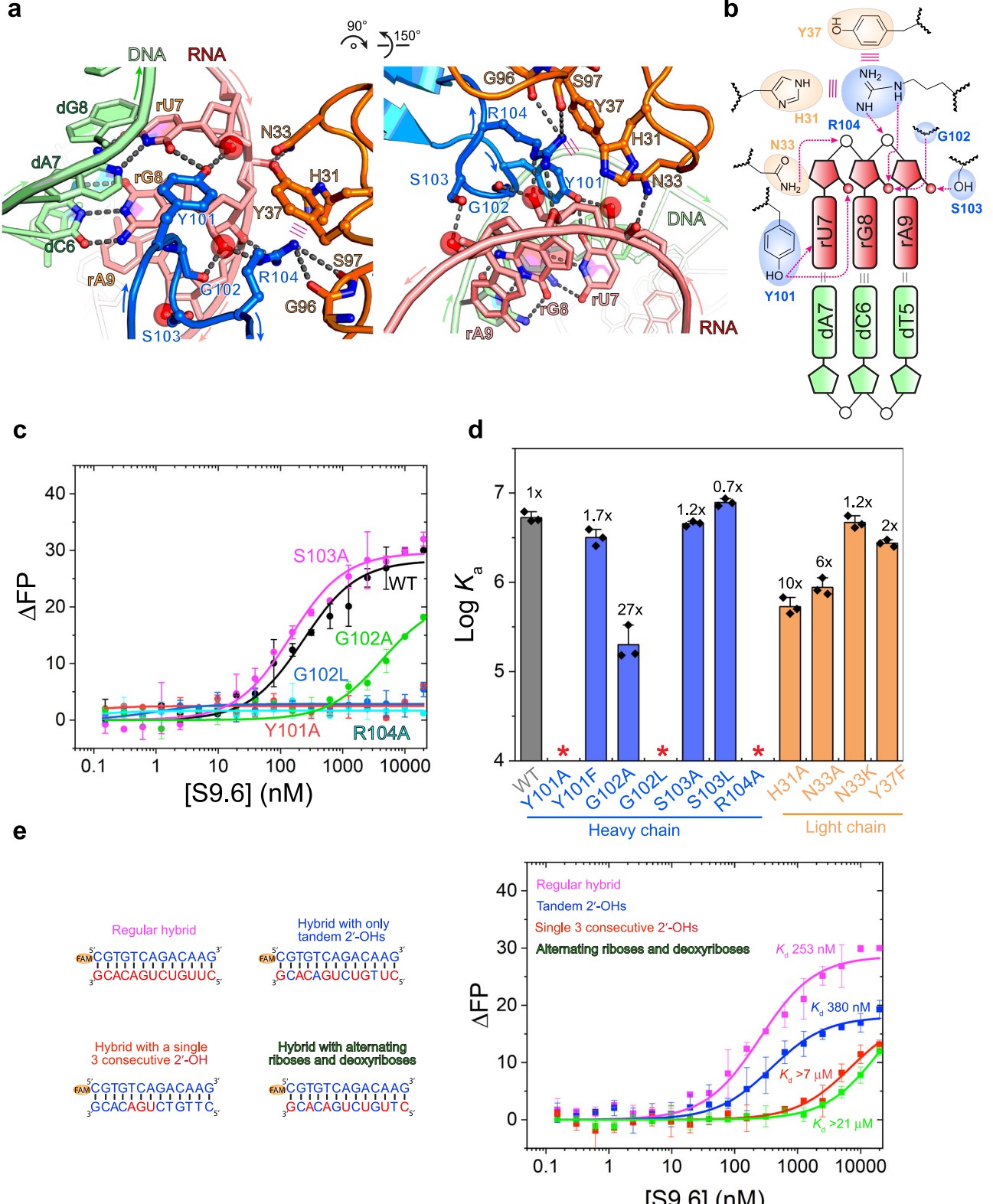

**Fig. 3 Recognition of the RNA strand by S9.6. a** Two views of the S9.6 interface with the RNA strand (red). Hydrogen bonds are shown as black dashed lines; stacking or cation–π interactions as parallel magenta lines. **b** Schematic representation of S9.6 interaction with the RNA strand. **c** Fluorescence polarization titration analysis of a 13-bp hybrid binding by WT S9.6 and S9.6 variants harboring mutations at the RNA-binding interface. **d** Effects of S9.6 mutations on hybrid binding, calculated from **c**. *: no significant binding detected. **e** Sequences of 13-bp double-stranded (ds) nucleic acids variants with different numbers of consecutive 2′-OHs used in fluorescence polarization titration analysis of WT S9.6 binding. Locations of FAM labels are also shown. **f** Fluorescence polarization titration of WT S9.6 to nucleic acids in **e**. Values are mean ± s.d. $n = 3$ biologically independent samples. Source data are provided as a Source Data file.

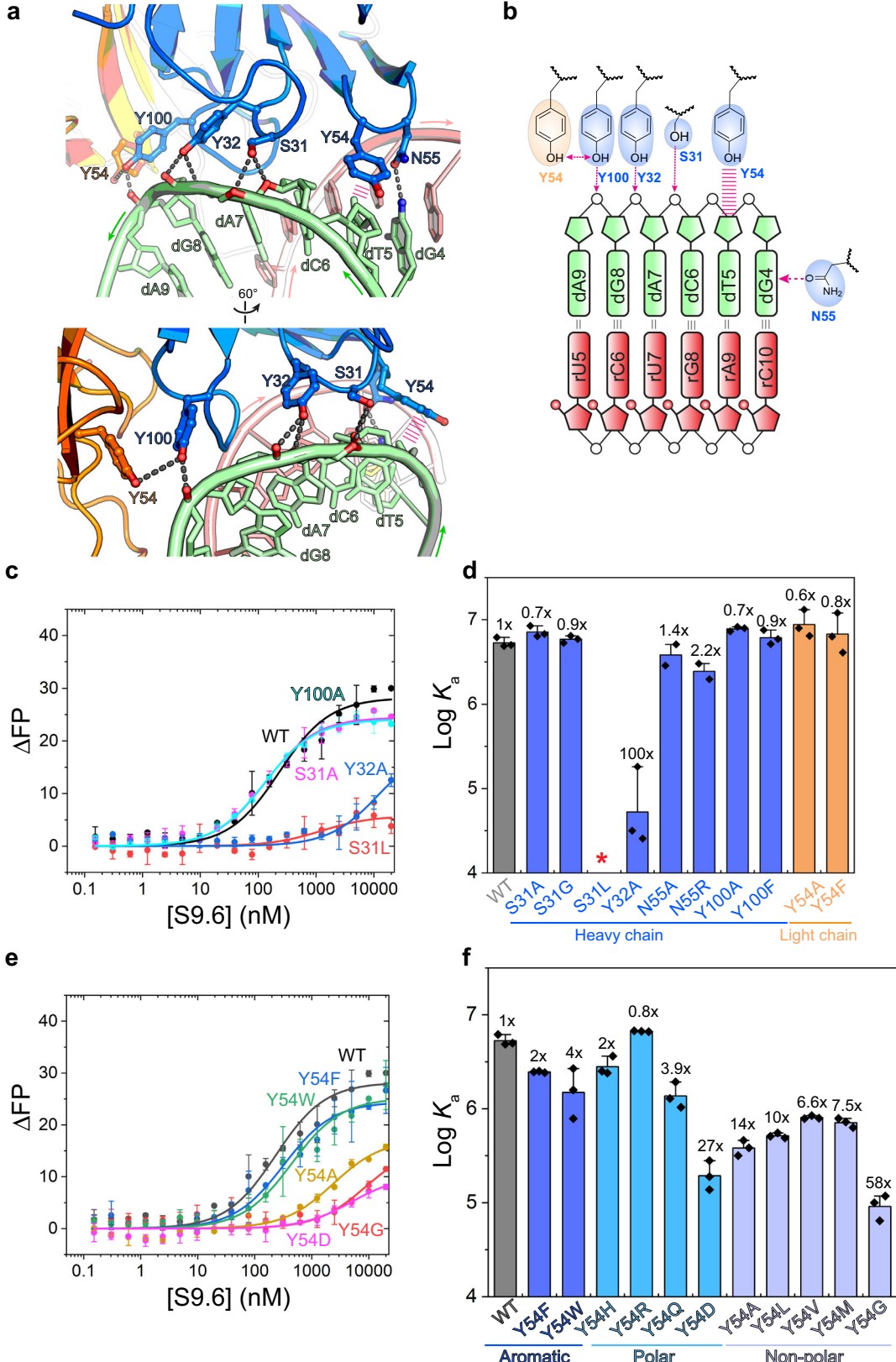

**Fig. 4 Recognition of the DNA strand by S9.6. a** Two views of the S9.6 interface with the DNA strand (green). Hydrogen bonds are shown as black dashed lines; stacking or cation–π interactions as parallel magenta lines. **b** Schematic representation of S9.6 interaction with the DNA strand. **c** Fluorescence polarization titration analysis of a 13-bp hybrid binding by WT S9.6 and S9.6 variants harboring mutations at the DNA-binding interface. **d** Effects of S9.6 mutations on hybrid binding, calculated from **c**. **e** Fluorescence polarization titration analysis of a 13-bp hybrid binding by WT S9.6 and S9.6 variants at the Y54 site. **f** Effects of Y54 mutations on hybrid binding, calculated from **e**. *: no significant binding detected. Values are mean ± s.d. $n = 3$ biologically independent samples. Source data are provided as a Source Data file.

substitution had a minor fourfold defect while the anionic Y54D led to a much larger 27-fold binding defect.

The particular Y54-dT5 sugar–π packing geometry may provide an anti-determinant against dsRNA binding, as a modeled ribose 2′-OH here protrudes perpendicularly towards the centroid of the hydrophobic Y54 benzene ring with a distance to the ring plane of 2.3 Å, and is thus expected to sterically disrupt the important sugar–π interaction. This contact is highly analogous to a DNA-selective stacking contact between deoxyribose and an indole ring of W221 of the basic protrusion of human RNase H1[45]. To test this notion, we introduced 2′-OH groups to either dT5 alone or a stretch of five DNA nucleotides centered at dT5. Curiously, S9.6 maintained normal binding to these chimeric constructs (Supplementary Fig. 2c). It is possible that S9.6 tolerated these 2′-OH groups by either binding at another site on the hybrid, shifting its binding register slightly to avoid the clash, or through a local CDR-H2 conformational change moving Y54 away.

Next, we explored whether the side chain identity at Y54 impacts the hybrid/dsRNA selectivity, by measuring dsRNA binding by the Y54 variants. Aromatic substitutions (Y54F and Y54W) and Y54H had little impact on either dsRNA or hybrid binding, thus maintaining the same selectivity (Supplementary Fig. 3o). Interestingly, polar substitutions Y54R and Y54Q substantially enhanced binding to dsRNA, likely through acquired polar interactions with the RNA phosphate backbone or 2′-OH (Supplementary Fig. 3p). As a result, they reduced hybrid/dsRNA selectivity. Most tested nonpolar residues in place of Y54 had little impact on dsRNA binding and hybrid/dsRNA selectivity. Together, these findings suggest that an aromatic or histidine residue at position 54 is a superior choice to a long-chain basic residue for specific hybrid binding, as it enhances DNA strand binding via a sugar–π interaction, without a concomitant increase of RNA binding conferred by a basic residue.

Lastly, at the upstream edge of the epitope, dG4 is recognized by N55 of CDR-H2 via its exocyclic 2-amino group in the minor groove (Fig. 4a and Supplementary Fig. 7). This nucleobase-specific contact to the DNA strand, like the Y101-rU7 contact to the RNA strand, is also dispensable for hybrid binding. This is evidenced by the negligible impact on affinity by both N55A, N55R substitutions and the swapping of the dG4-rC10 pair with d4C-rG10 (Fig. 4d and Supplementary Fig. 7).

In summary, the co-crystal structure reveals that S9.6 recognizes an extended segment of six nucleotides of the DNA strand, principally through Y32 recognition of two backbone phosphate oxygens and Y54 recognition of deoxyribose via a sugar–π interaction. There are additional peripheral contacts to the DNA strand that do not appear to contribute substantially to the binding. In contrast to the complete ablation of binding by single disruptions of the localized RNA contacts (e.g., Y101A and R104A), the more dispersed DNA contacts are individually less essential.

**Distinct effects of S9.6 and RNase H binding on the hybrid helical geometry.** The S9.6-bound hybrid adopts an overall A-form helical geometry, as do free hybrids and dsRNA (Fig. 5). By contrast, dsDNA takes B-form, with drastically different geometries in helical diameter, groove widths, depths, etc. In particular, the narrow, deep minor grooves in dsDNA (~6 Å wide and ~4 Å deep in B-form compared to ~11 Å wide and ~1 Å deep in A-form) likely prevents access by the shallow minor groove-binding S9.6. This incompatibility in helical geometry and the lack of 2′-OHs mediate emphatic rejection of dsDNA by S9.6 (Fig. 1). By contrast, the high degree of geometric similarities

between the hybrid and dsRNA drives significant cross-reactivity between the two types of duplexes, with S9.6 and RNase H family of proteins.

Although hybrid duplexes generally adopt A-form, essentially all their helical parameters are intermediate between dsDNA and dsRNA, closer to the latter[46]. The DNA and RNA strands within the hybrid duplex partially retain their B-form and A-form characters, respectively. The DNA strand backbone is conformationally more malleable compared to the more rigid RNA due to constraints imposed by the ribose 2′-OH. Indeed, the flexibility of the DNA strand is exploited by both bacterial and eukaryotic type-I RNases H for hybrid recognition[47–49]. Both enzymes distort the DNA strand and use a conserved "phosphate-binding pocket" to capture one conformationally constrained backbone phosphate, as a proxy to select for hybrids over dsRNA (Fig. 6). RNase H binding to the hybrid compresses the minor groove and widens the major groove, effectively driving the helical geometry towards B-form dsDNA (Fig. 5m). By contrast, S9.6 does not drastically alter the groove widths. The fact that S9.6 binding does not require or induce drastic conformational changes on either the hybrid or protein side reduces the entropic cost of binding and enables higher affinity, enthalpically driven interaction.

**Comparing and contrasting hybrid-recognition strategies employed by S9.6 and RNases H.** Most RNase H domains, as do many other dsDNA and dsRNA binding proteins, employ positive dipoles of α-helices to bind the minor groove and to position key residues[41]. However, the six CDRs of S9.6 facing the antigen are devoid of α-helices, contrasting with the Fab BL3–6 which recognizes an RNA hairpin loop using two short α-helices in its CDRs[50,51]. Instead, antibody CDRs are hypervariable loops presented from their β-sandwich framework of the conserved immunoglobulin fold, and frequently form β-hairpins and other internally stabilized loops (Fig. 2a–c). The S9.6 heavy chain fully utilizes its three CDRs to position three key aromatic residues and one crucial basic residue along the minor groove, from Y32 of CDR-H1, to Y54 of CDR-H2, and to Y101 and R104 of CDR-H3. This overabundance of aromatic residues, especially Tyr, is notable and also seen in other RNA- and protein-binding Fabs[52]. Indeed, aromatic residues are overwhelmingly used by RNases H to recognize the DNA strand (Supplementary Fig. 8). This is evidently driven, at least in part, by the need to recognize deoxyribose and reject ribose through ubiquitous sugar–π interactions[44].

Besides this direct recognition of deoxyribose over ribose by strategically placed aromatic or histidine residues, most RNases H have evolved another, indirect strategy, in which they select for a DNA strand by locally distorting its backbone into B-form and testing if it can kink and fit into a phosphate-binding pocket (Fig. 6)[45,47,49]. S9.6 has not developed such a deformability-based strategy. In contrast to the great lengths gone to recognize the DNA strand, detection of the rigid RNA strand is more straightforward. All known RNases H and the S9.6 antibody converge on making direct hydrogen bonds to 2′-OHs. Nonetheless, the number and distribution of these contacts vary, with RNases H1 and H3 catalytic domains each recognizing four successive 2′-OHs whereas S9.6 only requires two (Fig. 2c and Supplementary Fig. 8)[45,47,49].

**S9.6 exhibits strong preference for binding GC rich hybrids.** Despite its widespread usage to map R-loops in cells, S9.6's sequence specificity has not been systematically assessed. A recent report found large differences in S9.6 binding affinities to hybrids of different sequences with no clear pattern[32]. Our co-crystal structure identified two sequence-specific contacts to the

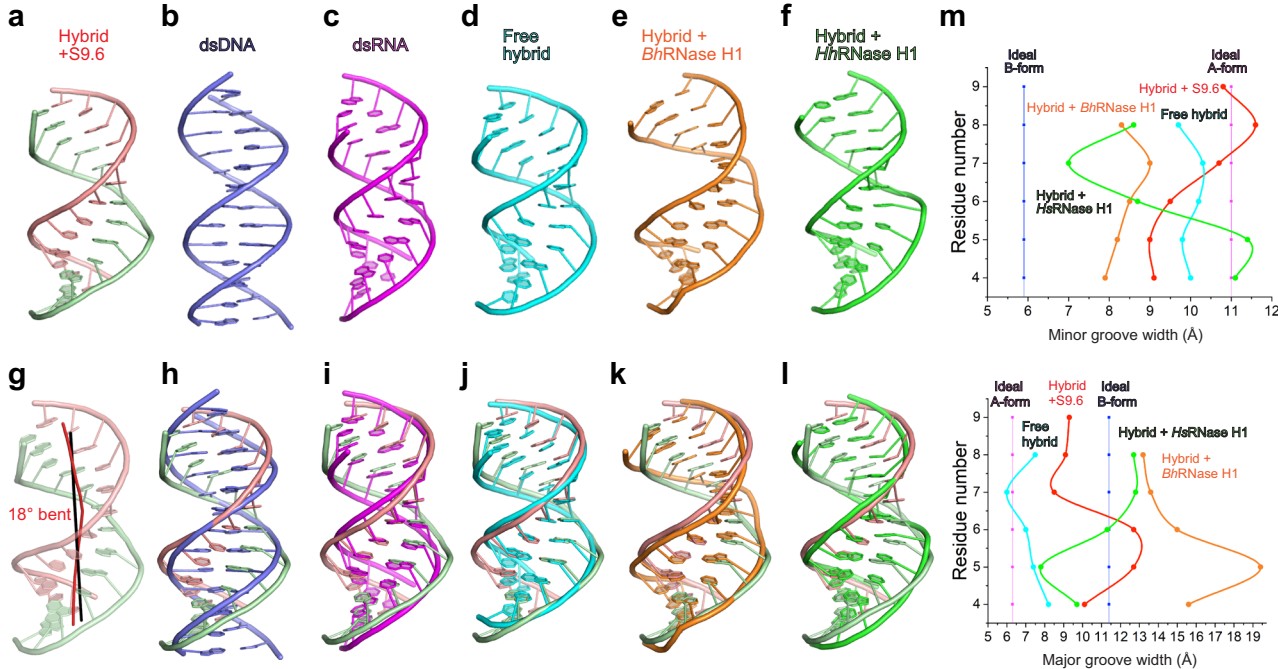

**Fig. 5 Effects of S9.6 and RNases H binding on hybrid helical geometry. a–f** Structural comparison between **a** DNA-RNA hybrid bound to S9.6 (RNA and DNA strands are in red and green), **b** Model dsDNA (blue), **c** Model dsRNA (magenta), **d** free hybrid (cyan, PDB 4WKJ[48]), **e** hybrid bound to *Bacillus halodurans* RNase H1 (orange, PDB 1ZBL[49]), and **f** hybrid bound to human RNase H1 (green, PDB 2QK9 (ref. [45])). **g** Helical trajectory of the S9.6-bound hybrid (red) compared to that of a model dsRNA (black). **h–l** Structural superposition of the S9.6-bound hybrid onto the structures above (**b–f**), with **h** dsDNA, **i** dsRNA, **j** free hybrid, **k** hybrid bound to *B. halodurans* RNase H1, and **l** hybrid bound to human RNase H1. **m** Minor (top) and major (bottom) groove widths for nucleic acids shown in (**a–f**). Source data are provided as a Source Data file.

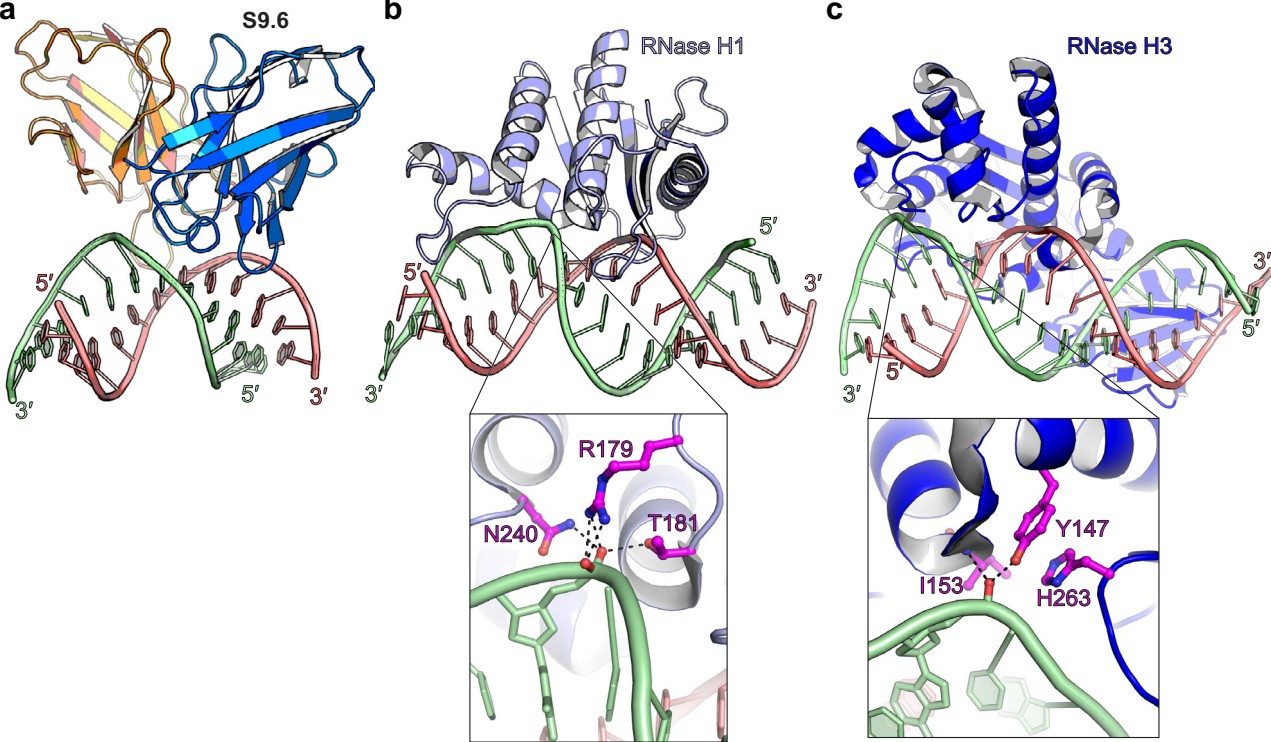

**Fig. 6 Comparison of hybrid recognition by S9.6 and RNases H. a** Hybrid binding by S9.6. **b** Hybrid binding by human RNase H1 with a zoomed-in view of the phosphate-binding pocket (PDB 2QK9 (ref. [45])). **c** Hybrid binding by *Thermovibrio ammonificans* RNase H3 with a zoomed-in view of the phosphate-binding pocket (PDB 4PY5 (ref. [81])).

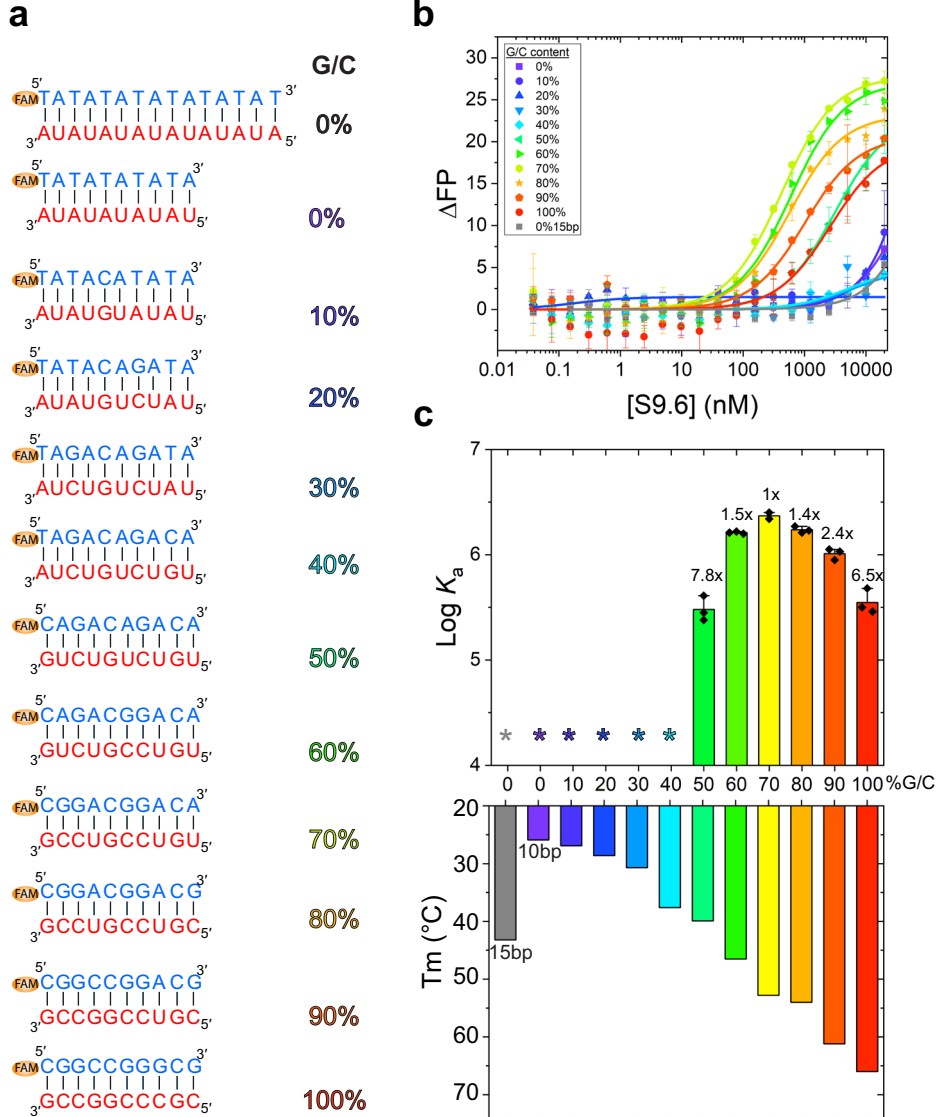

**Fig. 7 Effects of hybrid GC content on S9.6 binding. a** Sequences of 10-bp hybrids with increasing GC content (%) and a 15-bp hybrid with 0% GC content. **b** Fluorescence polarization titration of hybrids in **a** with S9.6. **c** Effect of GC content on S9.6 binding, calculated from **b** and melting temperatures ($T_m$) for each hybrid. *: no significant binding detected. Values are mean ± s.d. $n = 3$ biologically independent samples. Source data are provided as a Source Data file.

nucleobases of dG4 and rU7. However, when mutated, the absence of neither contact reduced S9.6 binding (Figs. 3, 4 and Supplementary Fig. 7).

To ask if S9.6 exhibits significant sequence preferences, we measured its binding to a series of eleven 10-bp hybrids with 10% increments in GC content. CD analyses indicated that these hybrids formed duplexes under our experimental conditions (Fig. 7 and Supplementary Fig. 9). We avoided long stretches of homotypic repeats such as poly-A. S9.6 exhibits a clear preference for GC-rich hybrids, with 70% GC content being optimal in this particular sequence context (Fig. 7b, c). For comparison, the 6-bp epitope on the hybrid in the crystal structure has 50% GC content. Only weak binding was observed when GC content was 40% or less (Fig. 7). As GC content strongly impacts the thermostability of short hybrids, we next asked if reduced hybrid stability may have been responsible for the lack of S9.6 binding. To this end, we measured S9.6 binding to 15-bp hybrid with 0% GC content. This stable hybrid has a $T_m$ of 43 °C, exhibits clear CD signatures of A-form duplexes, but had essentially no binding

to S9.6 (Fig. 7 and Supplementary Fig. 9). For comparison, the 50% GC 10-bp hybrid has a lower $T_m$ of 40 °C but is bound robustly by S9.6. Thus, the thermostability of hybrids is not a principal determinant of S9.6 binding and is not responsible for S9.6's inability to bind GC-less hybrids. These findings are further consistent with the original report that S9.6 bound poly (dT-rA) about 100-fold weaker than poly (dC-rI)[28] and a recent report that S9.6 did not strongly bind hybrids harboring long poly (dA-rU) segments[32]. Considering the mere 6-bp hybrid epitope (containing 3 GC pairs) required for S9.6 recognition, and a median R-loop length of 1.5 kb in cells, most R-loops are expected to harbor sufficient adjacent GC pairs for S9.6 recognition[4,29]. R-loop detection by S9.6 antibodies is further helped by the increased avidity from two spatially adjacent Fabs each recognizing a 6-bp hybrid. Indeed, a tiling microarray analysis using 60-nt long DNA probes and S9.6 antibodies to report hybridization to map the *Schizosaccharomyces pombe* transcriptome found that GC content had a small but positive correlation with the detected signal[53]. Overall, our findings, in conjunction with previous

studies, suggest that S9.6 exhibits a substantial intrinsic preference for GC-rich over AT-rich sequences, and has reduced ability to bind short, AU/AT-rich R-loops. A full accounting of the sequence preferences of S9.6, using high-coverage hybrid pools of known abundances and RNA-seq, will aid the interpretation of S9.6-based analyses of R-loop genomic distributions.

## Discussion

The chief findings of this work are: (a) S9.6 has an innate preference for binding DNA-RNA hybrids over dsRNA and dsDNA with a specificity comparable to RNases H, (b) S9.6 binds the hybrid minor groove and recognizes a compact, asymmetric epitope consisting of tandem RNA 2′-OHs and six DNA backbone linkages, and (c) S9.6 exhibits a substantial preference for GC-rich hybrids.

Our data provide quantitative assessments of the relative binding strengths of S9.6 for hybrids, dsRNA, and dsDNA in both bimolecular binding and competitive settings. In competitive settings, which approximate intracellular environments, unlabeled hybrids are more than 100 times better than dsRNA in competing for S9.6. However, the intracellular concentration of dsRNA far exceeds those of hybrids and R-loops, which overpowers the inherent preference of S9.6. As a result, most S9.6 immunofluorescence signal in cells is attributable to S9.6 binding to cellular dsRNA, especially to ribosomal RNA, instead of to hybrids[31]. Another factor that modulates S9.6 engagement in cells is competition from endogenous dsRNA- and hybrid-binding proteins. While exposed long dsRNA segments are generally avoided to prevent activation of antiviral immune sensors, S9.6 recognizes a much smaller epitope (6 bp) than most dsRNA sensors such as PKR, which needs at least 30 bp of dsRNA for activation[54,55], or MDA5 which binds ~500 bp[56]. Enhancing the hybrid/dsRNA selectivity by S9.6 is an inherently difficult task, due to their geometrical and chemical similarities and to a paucity of deployable anti-determinants for dsRNA binding. Our S9.6 structure shows that an anti-determinant for dsRNA rejection appears to be the sugar–π packing interaction by Y54 with the DNA strand. Conceivably, antibody engineering or selection may install additional such deoxyribose-specific contacts to better select against dsRNA. Our structure indicates that the light chain CDRs are promising regions, especially CDR-L2, to host additional affinity or specificity determinants, as they engage limited contacts with the hybrid despite their proximity to the minor groove. For instance, K55[L] and N58[L] from CDR-L2 are located only 7.5 and 6.9 Å away from the DNA strand. Thus, our structure provides a framework to design targeted S9.6 variant libraries to potentially enhance its hybrid/dsRNA selectivity.

Second, our data and previous studies collectively establish that S9.6 prefers to bind hybrids with medium and high GC content, with substantially weaker binding to 10-bp hybrids when GC content falls below 40% (Fig. 7). The exact reason underlying this GC preference remains unclear but may reflect sequence effects on helical geometry rather than sequence-specific contacts. The S9.6 structure reveals only two direct contacts to the nucleobases of dG4 and rU7 (Supplementary Fig. 7). The dG4 contact is a single hydrogen bond between its 2′-amino group and N55 side chain. This sole GC-specific contact is evidently unimportant for S9.6 binding, as evidenced by the lack of effects of the dG-rC swap with dC-rG, and mutating N55. These findings suggest that the GC preference is recognized indirectly, presumably through the local or global geometric features of the helix. While the RNA strand in hybrids generally maintains a rigid, A-form, C3′-endo conformation, the opposite DNA strand is much more flexible, which can assume A-like, B-like, or intermediate conformations

with various backbone and glycosidic dihedral angles, pseudorotation angles, and sugar puckers[46,57]. The mechanisms underlying how sequence and base composition exert drastic effects on helical geometry, stability, and deformability for dsDNA, dsRNA, and hybrids, are incompletely understood[46,57,58]. For instance, swapping the purine-rich and pyrimidine-rich strands in a 12-bp hybrid drastically alters its thermostability, with the hybrid containing a purine-rich RNA strand 7 kcal/mol more stable than its counterpart harboring a pyrimidine-rich RNA[59,60]. Similarly, a conserved polypurine tract (PPT) in the HIV-1 genome forms an unusually stable hybrid whose dsRNA-like, A-form geometry and repetitive, slippage-prone sequence escape HIV-1 reverse transcriptase RNase H cleavage, thereby leaving PPTs in place as primers of plus-strand DNA synthesis[61,62]. Dinucleotide steps such as tracts of repeating dinucleotides (such as AU) can induce significant helical bending in dsRNA[63]. Curiously, most reported free hybrid structures are bent (10–27°) while RNase H binding was suggested to straighten them (to 7–10°)[48]. For comparison, the S9.6-bound hybrid is bent by 18° compared to a model dsRNA (Fig. 5g). It is unknown if S9.6 induced the bend or the free hybrid was already bent in the first place. In either case, sequences that are not conducive to forming such bends may not be recognized by S9.6 with the same efficiency or affinity. Interestingly, RNases H also exhibits divergent and incompletely understood sequence preferences that depend on helical geometry and GC content[64]. Various DNA strand modifications that decrease its flexibility reduced RNase H cleavage, consistent with a requirement to distort the DNA strand to fit it into the phosphate-binding pocket[65]. Taken together, the hybrid duplex is a dynamic, polymorphic structure whose conformational flexibility and deformability are markedly impacted by its base composition and sequence, all of which may contribute to the observed sequence preferences by both S9.6 and RNases H. The short duplex epitope that S9.6 recognizes may have intensified its sequence dependency. Extending the epitope by daisy-chaining several S9.6 units together, such as in the form of scFv[30], can potentially reduce its apparent sequence dependency and produce more even binding to hybrids of different sequences.

The innate sequence preference of S9.6 has implications for interpreting genome-wide R-loop mapping data. Most such analyses so far have employed S9.6 immunoprecipitation and found that R-loops are enriched at regions of high GC content and GC skew[12,23] with a limited signal at AT-rich regions[2]. It could be that the R-loop-prone sequences and S9.6 preferences happen to align by coincidence. Alternatively, the GC preference of S9.6 may have introduced some degree of bias for GC-rich R-loops and underestimated AT-rich ones, which could be corrected or normalized when S9.6 sequence preferences become quantitatively known. Nonetheless, R-loop distributions mapped using S9.6 and non-S9.6 methods such as sodium bisulfite, various RNase H domains, or AID agree broadly, albeit with divergences[34,35,38,66,67]. Systematic, quantitative characterizations of the sequence specificities of S9.6 and RNases H are needed to accurately map R-loops genome-wide using them as hybrid detectors[64]. Besides sequences, posttranscriptional modifications such as m6A methylation further modulate R-loop formation and stabilities[68,69]. Since cytosine modifications markedly impact dsDNA flexibility[70], these and other nucleotide modifications are also expected to impact hybrid geometry and flexibility, and consequently S9.6 binding.

Despite the prevalence of naturally occurring nucleic acid antibodies in patients suffering from autoimmune diseases such as systemic lupus erythematosus[71] and antibodies widely used to detect nucleic acids in cells[72], relatively little is known about how antibodies recognize the nucleic acid structure and sequence[52]. Recently, phage display has enabled facile selection of novel

synthetic antibodies that bind RNA structures including ssRNA, hairpins, bulges, and junctions, thus expanding our knowledge of RNA recognition by antibodies[52,73]. The structural elucidation of S9.6 recognition of hybrids paves the way to examine other nucleic acid antibodies to understand their principles of recognition and basis of specificity, such as the recognition of long dsRNA by the J2 antibody[74], and of triplex DNA by the Jel 318 antibody[75]. Such knowledge will guide antibody engineering towards desirable traits and specificities. With the recent identification of hundreds of novel R-loop-binding human proteins[76], such as GADD45A[24], the hybrid-recognition strategies employed by S9.6 and revealed in this work may inform the understanding of these emerging R-loop-binding proteins.

## Methods

**Sequences and expression plasmids for S9.6 Fab**. The coding sequences for S9.6 HC and LC variable regions were known from prior work with the S9.6 scFv[30]. The remaining portions of the Fab HC and LC amino acid sequences were obtained initially from a preliminary crystal structure of the proteolytically-generated Fab and confirmed by MS/MS sequencing of the IgG and of Fab produced by digestion of the IgG. The DNA sequences were codon-optimized for expression in CHO cells and synthesized as gBlocks (Integrated DNA Technologies, Coralville, IA). The HC sequence was extended at the 3′ end to encode a sortase A signal ("SoSi", LPETGG) and hexa-histidine tag (His6). The ends of the gBlocks included XbaI (5′) and HindIII (3′) restriction sites with which the individual gBlocks were cloned into XbaI/HindIII cleaved expression vector pcDNA3.4 to generate pcDNA3.4-S9.6-Fab$_{LC}$ and pcDNA3.4-S9.6-Fab$_{HC}$-SoSi-His6. Amino acid substitutions were generated using the Q5 Site-Directed Mutagenesis Kit (NEB). Plasmid sequences and mutations were confirmed by Sanger sequencing (Psomagen, Rockville, MD). Amino acid sequences of the resulting S9.6 Fab HC and LC are given below, where signal sequences (removed during secretion) are underlined.

HC-SoSi-His6:
<u>MGWSCIILFLVATATGVHS</u>EVQLQQSGPELVKPGASVKMSCKASGYTFTS
YVM̲HWVKQKPGQGLEWIGFINLYNDGTKYNEKFKGKATLTSDKSSSTAYM
ELSSLTSKDSAVYYCARDYYGSRWFDYWGQGTTLTVSSAKTTAPSVYPLAPV
CGDTTGSSVTLGCLVKGYFPEPVTLTWNSGSLSSGVHTFPAVLQSDLYTLSS
SVTVTSSTWPSQSITCNVAHPASSTKVDKKISALPETGGGHHHHHH.
LC:
<u>MGWSCIILFLVATATGVHS</u>DVLMTQTPLSLPVSLGDQASISCRSSQSIV
HSNGNTYLEWYLQKPGQSPKLLIYKVSNRFSGVPDRFSGSGSGTDFTLKISR
VEAEDLGVYYCFQGSHVPYTFGGGTKLEIKRADAAPTVSIFPPSSEQLTSGG
ASVVCFLNNFYPKDINVKWKIDGSEVQNGVLNSWTDQDSKDSTYSMSSTL
TLTKDEYERHNSYTCEATHKTSTSPIVKSFNRNEC.

**Expression and purification of S9.6 Fab in ExpiCHO-S cells**. The ExpiCHO Expression System kit (ThermoFisher Scientific, Waltham, MA) was used according to manufacturer-recommended protocols with modifications. Briefly, ExpiCHO-S cells were grown in ExpiCHO expression medium in plain bottom Erlenmeyer flasks with vented screw caps at 37 °C and 8% CO$_2$ with continuous shaking at 120 rpm. ExpiCHO cells having viability >95% were transfected at a cell density of $5 \times 10^6$ cells/mL. Each 100-ml portion of expiCHO cells was transfected with 50 µg of each of HC and LC plasmids mixed with 8 mL of OptiPRO SFM and 320 µL of ExpiFectamine CHO reagent. These were combined and incubated for 1–5 min before addition to the cells. Following the "Max Titer" protocol, 600 µl ExpiCHO Enhancer and 16 ml of ExpiCHO Feed were added 24 h post transfection. The cells were moved to an incubator shaker kept at 32 °C and 5% CO$_2$, 120 rpm. An additional 16 ml ExpiCHO Feed was added on day 5 and the culture was harvested on days 10–12.

S9.6 Fab proteins were purified from culture supernatant using their C-terminal His6 tags. Following centrifugation to remove the expiCHO cells, the supernatant was incubated batchwise with gently rolling at 5 °C for at least 1 h with 5% (w/v) Ni-NTA agarose beads (Qiagen). This amount of Ni-NTA agarose exceeds that normally used because materials in the medium interfere, possibly by chelation of the nickel. The agarose beads were collected on a porous funnel or column, washed with 30 mM Imidazole, 200 mM NaCl, pH 7.0, and the Fab proteins were then eluted with 300 mM Imidazole, pH 7.0. The purified proteins were dialyzed in 10 mM Tris (pH 7.2) and further purified by size-exclusion chromatography on a Superdex 200 column equilibrated in 50 mM Tris-HCl pH 7.4, 150 mM NaCl and 2 mM MgCl$_2$. Protein purity was assessed by SDS-PAGE on 4–20% Tris-Glycine polyacrylamide gels (Invitrogen). For reduced samples, the S9.6 Fab proteins were treated with 1 mM dithiothreitol (DTT). To verify the presence of intended mutations, all Fab proteins were reduced by DTT and analyzed by mass spec. The intact HC and LC had masses within 1-2 mass units of the calculated values.

**Size-exclusion chromatography coupled with multi-angle light scattering (SEC-MALS)**. To assess binding stoichiometries of S9.6 with different types of nucleic acids, 40 µM of duplexes and 45 µM of S9.6 Fab were incubated for 10 min at room temperature in a buffer consisting of 25 mM Tris-HCl pH 7.5, 150 mM NaCl, and 2 mM MgCl$_2$, prior to injection onto a Superdex 200 Increase column on an Agilent HPLC system equilibrated in 25 mM Tris-HCl pH 7.5, 150 mM NaCl, and 2 mM MgCl$_2$. The HPLC system was coupled to a DAWN HELEOSII detector equipped with a quasi-elastic light scattering module and an Optilab T-rEX refractometer (Wyatt Technology). Data were analyzed using the ASTRA 7.3 software (Wyatt Technology Europe). All nucleic acids were purchased from Integrated DNA Technologies. All sequences of oligonucleotides are provided in Supplementary Data 1.

**Isothermal titration calorimetry (ITC)**. Nucleic acid duplexes were annealed in 25 mM Tris-HCl (pH 7.5) and 50 mM NaCl following the addition of 2 mM MgCl$_2$ at 65 °C and cooled to 4 °C with a ramp rate of 6 °C/min. The duplexes and S9.6 Fab were extensively exchanged into the same buffer containing 25 mM Tris-HCl (pH 7.5), 50 mM NaCl, and 2 mM MgCl$_2$ using Amicon Ultra Filter concentrators (Millipore). All ITC measurements were performed at 25 °C using a MicroCal iTC200 microcalorimeter (GE healthcare). About 20 µM of duplexes were used in the cell and titrated with 200 µM of S9.6. The raw ITC data were integrated using NITPIC and fit with SEDPHAT[77] to obtain the dissociation constants and thermodynamic parameters.

**Differential scanning calorimetry (DSC)**. DSC experiments were performed with 10 µM of 13-bp nucleic acid duplexes in 25 mM Tris-HCl pH 7.5, 50 mM NaCl and 2 mM MgCl$_2$ on a Malvern/GE VP-DSC instrument. The DSC instrument was equilibrated overnight with buffer in both sample and reference cells. The next day, the duplex was loaded in the sample cell and the DSC scan was recorded after a 60 min equilibration. The temperature range scanned was from 25 to 120 °C with a step of 1 °C min$^{-1}$. DSC data were corrected for instrument baselines and normalized for scan rate and duplex concentration. Data conversion and analysis were performed with Origin software (OriginLab Corporation, Northampton, MA, USA).

**Circular dichroism (CD)**. CD experiments were performed with 10 µM of DNA-RNA hybrids in 25 mM Tris-HCl pH 7.5, 50 mM NaCl and 2 mM MgCl$_2$ on an Applied Photophysics Chirascan$^{Tm}$ Q100 spectropolarimeter. The CD spectra were recorded from 320 to 195 nm using a 1 mm pathlength cell. The temperate range scanned from 20 to 97 °C with a step of 1 °C min$^{-1}$. The CD data were analyzed using the Global3 software provided with the instrument by Applied Photophysics.

**Binding measurements by fluorescence polarization (FP)**. About 5 nM of 5′-labeled duplexes with fluorescein were titrated with increasing amounts of S9.6 in a buffer consisting of 25 mM Tris-HCl (pH 7.5), 50 mM NaCl, and 2 mM MgCl$_2$ in a 96-well plate at 21 °C. FP values were measured in triplicates using a BMG CLARIOstar Plus microplate reader with excitation at 482 nm, emission at 530–540 nm, and LP (long pass) 504 dichroic filter setting. Changes in FP ($\Delta$FP) as a function of S9.6 concentrations were fit with the following equation to determine the apparent dissociation constant $K_d$:

$$y = \frac{\Delta \text{FP}_{max} * x}{K_d + x} \qquad (1)$$

**Competition experiments by fluorescence polarization**. About 10 µM of complexes were formed by mixing equimolar amounts of S9.6 Fab with unlabeled or 5′-FAM-labeled DNA-RNA hybrids and purified by gel filtration using a Superdex 200 Increase column on an AKTA Pure system equilibrated in 25 mM Tris-HCl pH 7.5, 50 mM NaCl, and 2 mM MgCl$_2$. About 5 nM of the purified labeled complex was quickly mixed with 150 nM of the purified unlabeled complex. Then increasing amounts of unlabeled duplexes were added to compete for S9.6 binding in a buffer composed of 25 mM Tris-HCl pH 7.5, 50 mM NaCl, and 2 mM MgCl$_2$, in a 96-well plate. FP values were recorded using the same settings as for the binding measurements above. Changes in FP as a function of competitor concentrations were fit to the following equation to determine the apparent half-maximal inhibitory concentration (IC$_{50}$):

$$y = \text{FP}_{final} + \frac{(\text{FP}_{initial} - \text{FP}_{final})}{(1 + 10^{(\log(x) - \log(\text{IC50}))})} \qquad (2)$$

**Crystallization, data collection, and structure determination**. For co-crystallization of S9.6 Fab bound to DNA-RNA hybrid, S9.6 was mixed with a 13-bp hybrid duplex in equimolar amounts and 7 mg/mL of the complex was mixed 1:1 with a reservoir solution consisting of 30% w/v PEG 1000 and 0.2 M sodium tartrate. Crystallization was performed at 20 °C by sitting-drop vapor diffusion. Rod-like crystals grew over 1 to 2 weeks to maximum dimensions of 120 µm$^3$ × 30 µm$^3$ × 30 µm$^3$. The crystals were cryoprotected in a synthetic mother liquor containing 32% PEG 1000, 0.2 M sodium tartrate, and 15% glycerol before

vitrification in liquid nitrogen. For crystallization of free S9.6 Fab, 10 mg/mL of the protein was mixed 1:1 with a reservoir solution consisting of 26% PEG 3350, 0.2 M ammonium sulfate, and 50 mM Bis-Tris pH 5.5. Cubic crystals appeared after 3 weeks with a maximum dimension of $40\ \mu m^3 \times 40\ \mu m^3 \times 40\ \mu m^3$. These crystals were cryoprotected with the addition of 15% glycerol to the synthetic mother liquor before vitrification in liquid nitrogen. All X-ray diffraction data were collected at the SER-CAT beamline 22-ID at the Advanced Photon Source (APS). X-ray diffraction data were indexed, integrated, and scaled using XDS via the xia2 package.

The co-crystal structure was solved by molecular replacement (MR) using PHASER[78]. The asymmetric unit contained one 13-bp DNA-RNA hybrid bound to a single S9.6 Fab. The heavy and light chain from PDB 3TT1 [https://www.rcsb.org/structure/3TT1] and an ideal 13-bp A-form dsRNA generated in Coot[79] were used as initial search models. The initial MR solution produced an overall TFZ (translation function Z-score) of 11.5 and an LLG (log-likelihood-gain) of 541. All 26 nucleotides forming the 13-bp DNA-RNA hybrid had well-defined electron density and were modeled. The free S9.6 structure was solved by molecular replacement using the S9.6 molecule from the co-crystal structure. The MR solution produced an overall TFZ of 26 and an LLG of 843 with three Fab molecules in the asymmetric unit. The structures were refined using phenix.refine[80] and iterative rounds of model building were performed using Coot. All crystallographic and refinement statistics are summarized in Supplementary Table 2.

**Reporting summary.** Further information on research design is available in the Nature Research Reporting Summary linked to this article.

## Data availability

The data that support this study are available from the corresponding authors upon reasonable request. The atomic coordinates and structure factor amplitudes for the free S9.6 Fab and S9.6 Fab in complex with a 13-bp hybrid duplex have been deposited at the Protein Data Bank under accession codes 7TQA [https://doi.org/10.2210/pdb7TQA/pdb] and 7TQB [https://doi.org/10.2210/pdb7TQB/pdb]. Previously released structural data used in the course of this study: 3TT1 [https://doi.org/10.2210/pdb3TT1/pdb], 4WKJ [https://doi.org/10.2210/pdb4WKJ/pdb], 1ZBL [https://doi.org/10.2210/pdb1ZBL/pdb], 2QK9 [https://doi.org/10.2210/pdb2QK9/pdb], 4PY5 [https://doi.org/10.2210/pdb4PY5/pdb]. Source data are provided with this paper.

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

## Acknowledgements

We thank I. Botos for computational support, G. Piszczek, D. Wu for support in bio-physical analyses, and R. Fattah, K. Singh, and L. Olano for support in S9.6 purification and characterization. X-ray diffraction data were collected at Southeast Regional Collaborative Access Team (SER-CAT) 22-ID beamline at the Advanced Photon Source of the Argonne National Laboratory, supported by the US Department of Energy under Contract No. W-31-109-Eng-38. This work was supported by the Intramural Research Program of the NIH, The National Institute of Diabetes and Digestive and Kidney Diseases (NIDDK) (ZIADK075136 to J.Z.), National Institute of Allergy and Infectious Diseases (NIAID), and an NIH Deputy Director for Intramural Research (DDIR) Challenge Award to J.Z. C.B.-N. is a recipient of an NIH Intramural AIDS Research Fellowship (IARF) and an NIDDK Nancy Nossal Fellowship Award.

## Author contributions

S.H.L. and J.Z. designed the work. C.B.-N. performed CD, DSC, FP, ITC, and SEC-MALS analyses. A.B. generated the mutant libraries. S.H.L. expressed all proteins. S.H.L. and C.B.-N. purified the proteins. D.N.G. performed initial experiments and sequenced S9.6 hybridoma cDNAs. C.B.-N. prepared the crystals, collected diffraction data, and determined the structures. C.B.-N. and J.Z. analyzed the structures. All authors contributed to manuscript preparation.

## Funding

## Competing interests

The authors declare no competing interests.
