## [Peer Review File · Nature Communications]

Structural basis of R-loop recognition by the S9.6 monoclonal antibodyReviewers' Comments:

Reviewer #1:

Remarks to the Author:

The manuscript submitted by Bou-Nader et al., titled "Structural basis of R-loop recognition by the S9.6 monoclonal antibody" presents a series of binding assays and a crystal structure of the Fab-hybrid complex to investigate the specificity of the antibody. While the complex structure is useful, along with some quantitative information on the binding affinities, more experiments are necessary to draw some of the major conclusions that the authors claim in the manuscript.

For the specificity of S9.6 in Fig 1, it is not clear whether different sequences were used for the duplexes. While the data looks consistent with the author's claims, if there was something different with one of the oligonucleotides (for dsRNA or dsDNA), this could result in disproportionate results. It would be helpful to show for example, that all three duplexes have the same stability (eg. melting temperature) as quality control. Furthermore, especially since this is one of the biggest points that the authors make--quantitative comparison of the binding affinities--it would be important to try different sequences. Is it possible that with a different sequence the preference for DNA:RNA hybrid change over RNA:RNA? While using 4 different methods look quite rigorous, if the same oligos were used, then the conclusions are drawn for just one particular situation which can be not representative.

If the interactions between the hybrid and S9.6 are not sequence-specific, how does this complex crystallize in a single register? This needs to be addressed more clearly.

The schematic diagrams to show the specific contacts are useful in Fig 2-4. However, the structure figures are not clear enough to understand the contacts. Thus, the reader is forced to take the arrows on the schematics without being able to evaluate the structures themselves. Especially for Fig 4a, it would help to find a way to make the interactions clearer.

The structure quality should be improved. There are too many clashes at this resolution. And for 3.1 Å, all disallowed residues should be fixed. The Ramachandron information should be included in the crystallography table. It would also help if the authors specified how many of the RNA/DNA nucleotides were modeled.

Only three 2'-OH contacts are observed, and the 3rd one that interacts with S103 looks inconsequential in the structure. And mutating S103 does not affect the binding. So it is possible that only 2 2'-OH is required for this recognition? Can the authors test this by not having 3 2'-OH's in a row?

If you only have 3 2'-OHs and the rest of the oligo was deoxyribose, would the Fab still bind?

If DNA is being recognized because of the absence of 2' OH near Y54, what happens if you do introduce 2'-OH there?

Y54G is a drastic change and might also interfere with N55 interactions. So it might be misleading to contrast it to the aromatic substitutions. What happens to Y54A? This would be important because there is no other non-aromatic substitution.

For the sequence preference of S9.6, the GC content seems to matter. However, is it possible that with higher GC content such a short oligo simply stays in the duplex form better, and thus the higher affinity for S9.6? This means that S9.6 simply prefers a stable duplex rather than higher GC content, and they are two separate questions. To test this the authors should show that the T_m of the lowest GC content is way above the experiment temperature. And they should test that a 10mer and a 15mer with the lowest GC content have similar affinities for S9.6. Otherwise, they might be seeing a T_m phenomenon, not "sequence preference"

Minor: Line 298 has a typo.

Reviewer #2:

Remarks to the Author:

This manuscript describes structural and functional characterization of the antigen binding fragment (Fab) derived from the S9.6 antibody. This antibody is used in immune-based methods to detect R-loops (stretches of DNA hybridized to RNA) within cells. Here the authors perform careful characterization of the binding affinity of the Fab for dsRNA, dsDNA, and RNA/DNA hybrids. They also obtain the crystal structures of Fab alone and Fab in complex with DNA/RNA hybrid. They find that the Fab is preorganized for binding and that recognition predominantly involves Y, S, and N residues from the heavy chain interacting with three residues of the RNA and five residues of the DNA. The authors describe details of the binding interface and test the importance of several interacting residues through mutagenesis. The authors further address the apparent GC rich sequence preference and what its possible origins could be. The description of the mode of binding to hybrids by other proteins, such as RNase H also adds value to the manuscript and puts the results gathered by the authors in a broader context.

Overall, the data and ensuing interpretation are rigorous, and the manuscript is presented in a clear, scholarly manner. Given that R-loops are relevant in some diseases and the S9.6 antibody is a promising tool for studies of R-loops, the topic overall has significance.

The following comments should be addressed:

Line 248: the authors state that "The particular Y54-dT5 sugar-pi packing geometry may have provided a chief anti-determinant against dsRNA binding, as a modelled ribose 2'-OH here protrudes perpendicularly. It seems this is a testable idea and considering all of the effort to test Fab mutations, the authors should test binding of the DNA-RNA hybrid bearing a 2'-OH at residue dT5.

Line 254 – Judging by this structure, the pi-sugar interaction between Y54 and dT5 is one of the most promising spots for improvement of specificity of S9.6 Fab. Unsurprisingly, mutations of Y54 to other aromatic residues did not change background binding to dsRNA, as these mutations did not change the nature of the interaction. However, it would be interesting to see if some branched residues with some potential for clashing, such as Ile, Leu, Val boosted the rejection of dsRNA by S9.6.

Line 297: The authors note how RNases H use alpha helices in nucleic acid recognition and that antibody CDR loops are beta hairpins. The authors may be interested to know Fab BL3-6, which binds the Class I ligase ribozyme, binds to an RNA hairpin, and its CDR-H3 has a short segment of alpha helix.

Line 309: the authors state that cellular proteins and antibodies have converged on this effective strategy to distinguish DNA and RNA, referring to the overabundance of aromatic residues, especially Tyr. However, antibodies, including those that bind proteins, generally have an overabundance of Tyr in their CDRs, and this has been attributed to the excellent molecular recognition characteristics of Tyr. Given this, it is not clear that this is a convergent property for binding nucleic acids.

For the binding studies of duplexes containing less GC rich character, it would be useful to know the T_m of those duplexes and whether the strands are in duplex form under the conditions of the binding experiments.

Minor:

Line 180: "assisted" does not seem to be the correct verb referring to "RNA strand".
Supplementary Fig. 3: label axis as accessible surface area that is buried.

Line 191, Fig. 3a – dA7 is often referenced in this paragraph, but it's not labeled in Fig. 3a. The described interactions between Y101 and dA7/rG8 are not well visible. It would be helpful if authors could label all relevant residues and perhaps include alternative, more detailed views that focus on described interactions (in SI?) to aid readers in interpretation of the structural features.

Line 192 – Y101-rG8 stacking is also not visible in Fig. 3a

Line 205 – _The authors say that G102L reduced binding likely due to changed geometry of the reverse turn. It seems possible, that another reason would be introducing clashes via the sidechain of Leu. A G102A mutation would likely exclude this possibility.

Line 208 – this interaction is obscured from view in Fig. 3a

Line 216 – do residues H31, Y37, Y101 interact with R104? It would be useful if it was shown on the structure.

Line 217 – similarly, G96 and S97 interactions with R104 are not presented in figures.

Figure 3b – side chains of Tyr and Arg are missing one CH₂ group (given how G102 and S103 are represented)

Figure 4b - Tyr, Asn are also missing one CH₂.

Supplementary Figure 2 - free Fab had 3 molecules per asymmetric unit. Were there some differences in CDRs between these 3 molecules? How many molecules per asymmetric unit were in Fab-hybrid complex?

Supplementary Figure 4i – the amine group of K204 does not seem to be positioned favorably for the Pi-cation bond formation with Y100, at least from the presented perspective, since it seems to face away from the aromatic ring

Reviewer #3:

Remarks to the Author:

Bou-Nader et al. present crystal structures of antigen-binding fragment of S6.9 antibody in apo form and in complex with an RNA/DNA hybrid. S9.6 is the most widely used tool to detect RNA/DNA hybrids. These are prevalent structures which are important for genome readout and maintenance playing both positive and detrimental roles. Therefore, their studies, in particular their sequencing, are important. All the tools that are currently used for RNA/DNA hybrid detection and isolation, including S9.6 antibody, have their serious shortcomings, so the development of new tools and optimization of the existing ones is an important area of research. This study is an important contribution towards this goal. It explains the mode of RNA/DNA recognition by S9.6 and provides the structural information which is required for the potential (but very challenging) rational improvement of the properties of this antibody. This work also broadens our understanding of molecular mechanisms of specific protein-RNA/DNA binding. It reinforces the notion that RNA/DNA recognition by proteins mostly relies on the interactions with 2'-OH groups of the RNA strand and stacking of aromatic side chains with ribose rings of the DNA strand.

This study is technically sound. Based on the validation reports the structures are of good quality. The presentation of the results and the figures are clear. The structural data are verified in

mutagenesis/binding experiments. The discussion of the results is very thorough and thoughtful.

The feature that distinguished S9.6 from other RNA/DNA-binding proteins is the fact that the IgG form of the antibody has two hybrid-binding sites. Based on the presented structural data, is it possible to model what would be the minimal length of a hybrid that can engage at both Fabs? One would assume that when such a length is reached the affinity of IgG for the hybrid would be dramatically increased.

What is puzzling is that the structure does not fully explain the RNA/DNA sequence preference of S9.6 which had been reported before and which is very apparent in the binding experiments presented in this work. This mostly likely relies on the dynamic properties of the nucleic acid and could be further explored (beyond the scope of the current work) using molecular dynamics simulations followed by biochemical studies.

In summary, this is a solid and important study and this reviewer does not find any weaknesses of this work.

Minor:

Line 69: Should be "senataxin"

Line 377: Should be "...provides a framework..."

Point-by-point responses to reviewer comments:

Reviewer #1 (Remarks to the Author):

The manuscript submitted by Bou-Nader et al., titled “Structural basis of R-loop recognition by the S9.6 monoclonal antibody” presents a series of binding assays and a crystal structure of the Fab-hybrid complex to investigate the specificity of the antibody. While the complex structure is useful, along with some quantitative information on the binding affinities, more experiments are necessary to draw some of the major conclusions that the authors claim in the manuscript.

A: We thank the reviewer for their positive and constructive feedback. As suggested, we have added substantive new experiments which expanded and corroborated the major conclusions. Specifically, we have added two additional techniques, differential scanning calorimetry (DSC) and circular dichroism (CD), to analyze the stability of the duplexes. We have generated 8 new S9.6 variants and added a number of new duplex constructs with various ribose/deoxyribose arrangements, to further examine the S9.6-hybrid interface and binding determinants both on S9.6 and on the hybrid. These are detailed below.

For the specificity of S9.6 in Fig 1, it is not clear whether different sequences were used for the duplexes. While the data looks consistent with the author’s claims, if there was something different with one of the oligonucleotides (for dsRNA or dsDNA), this could result in disproportionate results. It would be helpful to show for example, that all three duplexes have the same stability (eg. melting temperature) as quality control.

A: As shown in Fig 1a and mentioned in the text line 104, ‘the same length and sequence’ of dsRNA, dsDNA and DNA-RNA hybrid duplexes were used to assess S9.6 binding. We have edited the text to make it more clear that the sequences are identical. As suggested, we measured the melting temperatures of the three duplexes using DSC to ensure that they are all stable at room temp (T_m : 55-72°C, Fig. S1), and also obtained their CD spectra to show that they are indeed double-stranded under the binding conditions.

We added a new paragraph:

“First, we verified that these 13-bp nucleic acid duplexes are stable at room temperature (~21 °C) using circular dichroism (CD). CD spectra of all three assemblies exhibited signature bands of duplex nucleic acids at ~209, 262 and 280 nm (Supplementary Fig. 1), and showed that the hybrid structure possesses characteristics of both dsDNA and dsRNA and is closer to dsRNA, as reported previously³⁹. Further, we measured the thermostability of these duplexes by differential scanning calorimetry (DSC), which produced T_m values of 58, 55 and 72 °C for dsDNA, hybrid and dsRNA, respectively. Thus, both CD and DSC analyses confirm that the three 13-bp duplex nucleic acids of the particular sequence used are stable at 21 °C.”

Furthermore, especially since this is one of the biggest points that the authors make--quantitative comparison of the binding affinities--it would be important to try different sequences. Is it possible that with a different sequence the preference for DNA:RNA hybrid change over RNA:RNA? While using 4 different methods look quite rigorous, if the same oligos were used, then the conclusions are drawn for just one particular situation which can be not representative.

A: We appreciate the reviewer bringing up this important point. As suggested, we expanded our initial analysis to include two additional sequences derived from natural R-loops identified at the human *FUS* locus (PMID 31679819) and the beta-actin terminator region (PMID 32105733). As shown in Fig 1d & S2, S9.6 binds preferably to hybrids of all three distinct sequences while no interaction with dsDNA was detected. The K_d for the three hybrids ranges from 78 to 203 nM and from 1.5 to 5.4 μM for the dsRNAs of the same sequences. This shows that across at least

three distinct sequences, including at natural R-loop sites, S9.6 exhibits significant and variable binding preferences for hybrids over dsRNA, and reject dsDNA.

We now state:

'To ask if this binding preference of S9.6 is conserved towards different hybrid sequences, we measured S9.6 binding to two naturally occurring R-loop sequences at the FUS locus and β -actin terminator. In both cases, similar S9.6 preferences were observed (Supplementary Fig. 2), suggesting that S9.6 has a general propensity to preferably bind hybrids over dsRNA and essentially no affinity for dsDNA.'

If the interactions between the hybrid and S9.6 are not sequence-specific, how does this complex crystallize in a single register? This needs to be addressed more clearly.

A: The S9.6 epitope spans ~ 6 bp hybrid. On a relatively short 13-bp hybrid, to maximize the binding interface and favorable enthalpy, S9.6 is more likely to bind near the middle section than at the ends. The width of the Fab is approximately the same as the length of a 13-bp hybrid (see right Fig on crystal packing). It appears that concurrent Fab-Fab packing and adjacent hybrid end-to-end stacking may have normalized the binding register in the co-crystals. We did also attempt hybrids of other lengths, which failed to produce crystals, potentially due to their more heterogeneous, non-normalized registers and incompatibility between the hybrid packing and the Fab packing. Although it is clear from genome-wide mapping analyses and our more limited testing that S9.6 can bind many different sequences, it may still have some sequence preferences, such as G/C content and beyond. Thus, we can't exclude the possibility that most S9.6 molecules preferably bound a particular site on the hybrid, and this single-register complex happens to pack without substantial lateral sliding or register normalization. Other possible contributing factors to the observed unified register include the curvature and groove widths of the hybrid as a function of the sequence.

The schematic diagrams to show the specific contacts are useful in Fig 2-4. However, the structure figures are not clear enough to understand the contacts. Thus, the reader is forced to take the arrows on the schematics without being able to evaluate the structures themselves. Especially for Fig 4a, it would help to find a way to make the interactions clearer.

A: As suggested, we have edited current viewing perspectives and added orthogonal views to both Fig. 3a and Fig. 4a, to more clearly illustrate the interactions.

The structure quality should be improved. There are too many clashes at this resolution. And for 3.1 Å, all disallowed residues should be fixed. The Ramachandran information should be included in the crystallography table. It would also help if the authors specified how many of the RNA/DNA nucleotides were modeled.

A: We have corrected the PDB models as suggested, added the Ramachandran information in the crystallographic statistics table, and specified the numbers of modeled nucleotides both in Methods and in the crystallographic statistics table. All 26 nucleotides of the 13-bp hybrid were visible and modeled.

Only three 2'-OH contacts are observed, and the 3rd one that interacts with S103 looks inconsequential in the structure. And mutating S103 does not affect the binding. So it is possible that only 2 2'-OH is required for this recognition? Can the authors test this by not having 3 2'-OH's in a row?

A: We thank the reviewer for this excellent suggestion. Indeed, we did exactly this and found that 2 consecutive 2'-OHs are sufficient for binding and 3 is not needed (only ~50% higher K_d , Fig. 3e). This is fully consistent with the lack of a S103A phenotype. We further found that tandem 2'-OHs are also required for binding, as an alternating ribose/deoxyribose arrangement in the RNA strand bound poorly ($K_d > 7 \mu\text{M}$). Together, tandem 2'-OHs seem both necessary and sufficient for S9.6 binding, given that they are in an A-form duplex (see below).

If you only have 3 2'-OHs and the rest of the oligo was deoxyribose, would the Fab still bind?

A: No. We made such a construct and it did not substantially bind ($K_d > 21 \mu\text{M}$, Fig. 3e). To find out why, we measured its CD spectra and found this duplex to be mostly B-form, since most residues are now DNA.

If DNA is being recognized because of the absence of 2' OH near Y54, what happens if you do introduce 2'-OH there?

A: As suggested, we inserted 2'-OHs at dT5 and further at 5 DNA nucleotides centered at dT5. Curiously, we did not observe binding defects to these chimeras (Fig. S2c). There could be multiple explanations to this observation. S9.6 may bind at another site away from the dT5 region, or laterally slide a small distance to avoid the 2'-OH, or employ a local protein conformational change to avoid clashing with the 2'-OH. Regardless, it appears reasonable to conclude that steric clash with a single 2'-OH is insufficient to drive the bulk of the observed hybrid/dsRNA selectivity. More likely, it is the conformation or deformability of the hybrid that is the chief determinant. Thus, we removed the "chief" descriptor from the text and added this discussion to the text.

Y54G is a drastic change and might also interfere with N55 interactions. So it might be misleading to contrast it to the aromatic substitutions. What happens to Y54A? This would be important because there is no other non-aromatic substitution.

A: We thank the reviewer for their excellent suggestion, and have tested a wider range of residues at Y54, including Y54A, to better understand the nature of the interaction and also potentially to improve hybrid/dsRNA selectivity of S9.6 (Fig. 4f). We found that hydrophobic and/or branched residues at Y54 (Leu, Val, Met, Ala tested) generally reduced the binding to hybrids (by 7-14 fold) but not to dsRNA (Fig. S3m,q). As a result, these actually decreased the hybrid selectivity. We also tested 3 more polar residues (cationic, neutral, and anionic), and found that Y54R enhanced binding to both hybrids and dsRNA, Y54Q reduced hybrid binding but increased dsRNA binding, and Y54D barely bound the hybrids. These phenotypes are consistent with a newly created electrostatic interaction with the DNA backbone. The findings suggest that introducing an electrostatic interaction instead of the sugar- π interaction at Y54 generally worsens the hybrid/dsRNA selectivity as well. Taken together, an aromatic or histidine residue at position 54 appears to already be the best choice available, at least among the 12 tested side chains. These new data have been added to the manuscript .

For the sequence preference of S9.6, the GC content seems to matter. However, is it possible that with higher GC content such a short oligo simply stays in the duplex form better, and thus the higher affinity for S9.6? This means that S9.6 simply prefers a stable duplex rather than higher GC content, and they are two separate questions. To test this the authors should show that the T_m of the lowest GC content is way above the experiment temperature.

A: As suggested, we measured CD spectra for each of the 10-bp duplexes and monitored their unfolding transitions with increasing temperature to determine their T_m (Fig. S9). The CD spectra at 21°C shows that all hybrids formed structures close to an A-form helix. As the temperature was raised, the positive peak at ~260-265 nm gradually decreased and shifted consistent with unfolding of the duplex. The melting temperature for all 10-bp duplexes are at least 5°C higher than the experimental temperature of 21°C suggesting that the double-stranded hybrids were formed under the binding conditions. For the GC-less hybrid with the lowest T_m of 26 °C, a global fit of the CD spectra across all temperatures showed that ~80% of the nucleic acids were in duplexed form (Fig. S9a). We then further corroborated this below, by testing a 15-bp GC-less hybrid as suggested.

And they should test that a 10mer and a 15 mer with the lowest GC content have similar affinities for S9.6. Otherwise, they might be seeing a T_m phenomenon, not “sequence preference”

A: We appreciate this helpful suggestion, to tease apart potential effects on thermostability *versus* sequence preference. As proposed, we measured the stability and binding of a 15-bp GC-less hybrid. Despite a robust T_m of 43 °C (Fig. S9I), this 15-bp hybrid exhibited essentially no binding (Fig. 7b, gray squares). This T_m is evidently sufficient for binding, as the 50% G/C 10-bp hybrid which bound S9.6 robustly had a lower T_m of 40°C. We have added these important new data to Fig. 7. Our findings are consistent with two previous reports that S9.6 only weakly associated with GC-less hybrids longer than 10-bp (PMIDs: 2422282, 28594954, refs #28 & #32).

Minor: Line 298 has a typo.
Fixed.

Reviewer #2 (Remarks to the Author):

This manuscript describes structural and functional characterization of the antigen binding fragment (Fab) derived from the S9.6 antibody. This antibody is used in immune-based methods to detect R-loops (stretches of DNA hybridized to RNA) within cells. Here the authors perform careful characterization of the binding affinity of the Fab for dsRNA, dsDNA, and RNA/DNA hybrids. They also obtain the crystal structures of Fab alone and Fab in complex with DNA/RNA hybrid. They find that the Fab is preorganized for binding and that recognition predominantly involves Y, S, and N residues from the heavy chain interacting with three residues of the RNA and five residues of the DNA. The authors describe details of the binding interface and test the importance of several interacting residues through mutagenesis. The authors further address the apparent GC rich sequence preference and what its possible origins could be. The description of the mode of binding to hybrids by other proteins, such as RNase H also adds value to the manuscript and puts the results gathered by the authors in a broader context.

Overall, the data and ensuing interpretation are rigorous, and the manuscript is presented in a clear, scholarly manner. Given that R-loops are relevant in some diseases and the S9.6 antibody is a promising tool for studies of R-loops, the topic overall has significance.

We thank the reviewer for their favorable assessment and constructive and detailed suggestions.

The following comments should be addressed:

Line 248: the authors state that “The particular Y54-dT5 sugar-pi packing geometry may have provided a chief anti-determinant against dsRNA binding, as a modelled ribose 2'-OH here protrudes perpendicularly. It seems this is a testable idea and considering all of the effort to test Fab mutations, the authors should test binding of the DNA-RNA hybrid bearing a 2'-OH at residue dT5.

A: As suggested, we inserted 2'-OHs at dT5 and further at 5 DNA nucleotides centered at dT5. Curiously, we did not observe binding defects to these chimeras (Fig. S2c). There could be multiple explanations to this observation. S9.6 may bind at another site away from the dT5 region, or laterally slide a small distance to avoid the 2'-OH, or employ a local protein conformational change to avoid clashing with the 2'-OH. Regardless, it appears reasonable to conclude that steric clash with a single 2'-OH is insufficient to drive the bulk of the observed hybrid/dsRNA selectivity. More likely, it is the conformation or deformability of the hybrid that is the chief determinant. Thus, we removed the “chief” descriptor from the text.

Line 254 – Judging by this structure, the pi-sugar interaction between Y54 and dT5 is one of the most promising spots for improvement of specificity of S9.6 Fab. Unsurprisingly, mutations of Y54 to other aromatic residues did not change background binding to dsRNA, as these mutations did not change the nature of the interaction. However, it would be interesting to see if some branched residues with some potential for clashing, such as Ile, Leu, Val boosted the rejection of dsRNA by S9.6.

A: We thank the reviewer for their excellent suggestion, and have tested a wider range of residues at Y54, to better understand the nature of the interaction and also potentially to improve hybrid/dsRNA selectivity of S9.6 (Fig. 4f). We found that hydrophobic and/or branched residues at Y54 (Leu, Val, Met, Ala tested) generally reduced the binding to hybrids (by 7-14 fold) but not to dsRNA (Fig. S3m-q). As a result, these actually decreased the hybrid selectivity. We went a step further to also test 3 more polar residues (cationic, neutral, and anionic), and found that Y54R enhanced binding to both hybrids and dsRNA, Y54Q reduced hybrid binding but increased dsRNA binding, and Y54D barely bound the hybrids. These phenotypes are

consistent with a newly created electrostatic interaction with the DNA backbone. The findings suggest that introducing an electrostatic interaction instead of the sugar-pi interaction at Y54 generally worsens the hybrid/dsRNA selectivity as well. Taken together, an aromatic (or histidine) residue at position 54 appears to already be the best choice available, at least among the 12 tested side chains. More extensive engineering efforts, such as combinatorial mutations or main chain alterations, seem necessary to further improve the hybrid/dsRNA selectivity of S9.6. We also suggest that CDR-L2 engineering may be another path towards this improvement.

Line 297: The authors note how RNases H use alpha helices in nucleic acid recognition and that antibody CDR loops are beta hairpins. The authors may be interested to know Fab BL3-6, which binds the Class I ligase ribozyme, binds to an RNA hairpin, and its CDR-H3 has a short segment of alpha helix.

A: We thank the reviewer for suggesting this useful information and have incorporated it into the text. We now state:

“...contrasting with the Fab BL3-6 which recognizes an RNA hairpin loop using two short alpha helices in its CDRs.”

Line 309: the authors state that cellular proteins and antibodies have converged on this effective strategy to distinguish DNA and RNA, referring to the overabundance of aromatic residues, especially Tyr. However, antibodies, including those that bind proteins, generally have an overabundance of Tyr in their CDRs, and this has been attributed to the excellent molecular recognition characteristics of Tyr. Given this, it is not clear that this is a convergent property for binding nucleic acids.

A: We agree and have removed this statement.

For the binding studies of duplexes containing less GC rich character, it would be useful to know the T_m of those duplexes and whether the strands are in duplex form under the conditions of the binding experiments.

A: Indeed. As suggested, we measured the T_m of the duplexes using DSC and CD. Both analyses suggest that the duplexes used in the study are stable under the conditions of the binding experiment. We did note that the GC-less 10-bp hybrid had a relatively low T_m of 26°C, ~5 degrees above the binding experiment. So we also tested a longer 15-bp version as suggested by Reviewer #1, which is much more stable with a T_m of 43°C. Neither hybrid bound S9.6 (Fig. 7). For comparison, the 50% G/C 10-bp hybrid had a T_m of 40°C and bound S9.6 robustly. These findings suggest that the absence of GC pairs, not thermostability, is primarily responsible for the lack of S9.6 binding.

Minor:

Line 180: “assisted” does not seem to be the correct verb referring to “RNA strand”.

Corrected to say “...strand is recognized principally by CDR-H3, **which is** assisted by CDR-H1...”

Supplementary Fig. 3: label axis as accessible surface area that is buried.

Fixed.

Line 191, Fig. 3a – dA7 is often referenced in this paragraph, but it's not labeled in Fig. 3a. The described interactions between Y101 and dA7/rG8 are not well visible. It would be helpful if authors could label all relevant residues and perhaps include alternative, more detailed views that focus on described interactions (in SI?) to aid readers in interpretation of the structural features.

A: As suggested, we replaced Fig. 3a with two orthogonal views of the interface and labelled all relevant residues. dA7 and rG8 are now clearly visible in the left panel of Fig. 3a.

a

Line 192 – Y101-rG8 stacking is also not visible in Fig. 3a

A: New Fig. 3a left panel now visualizes this possible stacking, shown above.

Line 205 – The authors say that G102L reduced binding likely due to changed geometry of the reverse turn. It seems possible, that another reason would be introducing clashes via the sidechain of Leu. A G102A mutation would likely exclude this possibility.

A: Yes, we agree with this dual-effect interpretation. As suggested, we generated G102A and found a 27-fold defect as opposed to the total loss of G102L (Fig 3d). This is indeed fully consistent with backbone geometry as well as steric conflict playing a role in the dramatic defect of G102L.

Line 208 – this interaction is obscured from view in Fig. 3a

A: In the new Fig. 3a right panel, S103 contact with the 2'-OH of rA9 is now in the foreground on the left, shown above.

Line 216 – do residues H31, Y37, Y101 interact with R104? It would be useful if it was shown on the structure.

A: Y37 forms a cation- π interaction with R104, which is now highlighted by magenta parallel lines in both panels of the new Fig. 3a above. H31 appears just a tad too far to directly interact with R104, but may help constrain its long side chain. Y101 is too far to contact R104, which is visualized in the left panel of Fig. 3a.

Line 217 – similarly, G96 and S97 interactions with R104 are not presented in figures.

A: In the new Fig. 3a, both panels now show G96 and S97 interactions with R104, shown above.

Figure 3b – side chains of Tyr and Arg are missing one CH2 group (given how G102 and S103 are represented)

Fixed

Figure 4b - Tyr, Asn are also missing one CH2.

Fixed

Supplementary Figure 2 - free Fab had 3 molecules per asymmetric unit. Were there some

differences in CDRs between these 3 molecules? How many molecules per asymmetric unit were in Fab-hybrid complex?

A: The CDRs of the 3 molecules of free Fab are highly similar (RMSD <1 Å). Detailed pair-wise analyses can be found in Fig S6a-c. One Fab molecule was present per asymmetric unit in the co-crystal structure.

Supplementary Figure 4i – the amine group of K204 does not seem to be positioned favorably for the Pi-cation bond formation with Y100, at least from the presented perspective, since it seems to face away from the aromatic ring

A: We thank the reviewer for catching this mis-annotation, and have removed this notation. The Y100-K204' contact seems hydrophobic in nature, as the aromatic ring of Y100 is ~3.9 Å from the methylene groups of K204'.

Reviewer #3 (Remarks to the Author):

Bou-Nader et al. present crystal structures of antigen-binding fragment of S6.9 antibody in apo form and in complex with an RNA/DNA hybrid. S9.6 is the most widely used tool to detect RNA/DNA hybrids. These are prevalent structures which are important for genome readout and maintenance playing both positive and detrimental roles. Therefore, their studies, in particular their sequencing, are important. All the tools that are currently used for RNA/DNA hybrid detection and isolation, including S9.6 antibody, have their serious shortcomings, so the development of new tools and optimization of the existing ones is an important area of research. This study is an important contribution towards this goal. It explains the mode of RNA/DNA recognition by S9.6 and provides the structural information which is required for the potential (but very challenging) rational improvement of the properties of this antibody. This work also broadens our understanding of molecular mechanisms of specific protein-RNA/DNA binding. It reinforces the notion that RNA/DNA recognition by proteins mostly relies on the interactions with 2'-OH groups of the RNA strand and stacking of aromatic side chains with ribose rings of the DNA strand.

This study is technically sound. Based on the validation reports the structures are of good quality. The presentation of the results and the figures are clear. The structural data are verified in mutagenesis/binding experiments. The discussion of the results is very thorough and thoughtful.

We thank the reviewer for their favorable assessments and constructive suggestions.

The feature that distinguished S9.6 from other RNA/DNA-binding proteins is the fact that the IgG form of the antibody has two hybrid-binding sites. Based on the presented structural data, is it possible to model what would be the minimal length of a hybrid that can engage at both Fabs? One would assume that when such a length is reached the affinity of IgG for the hybrid would be dramatically increased.

A: In the context of IgGs, the flexible hinge region that connects the Fab to the Fc region allows the peripheral Fabs to pivot (PMID: 7755903, 2995045, 6506711).

These pivoting motions produce a wide range of conformations and varied architectures of the IgG. This flexibility suggests that the modeling would have limited precision.

Nonetheless, we attempted

rudimentary modeling, by superposing two copies of our co-crystal structure onto the Fabs of an intact anti-phenobarbital IgG1 antibody structure (PDB 1IGY). The Fabs superposed well with an overall RMSD of 1.1 Å. To connect the two bound 13-bp hybrids, we then superposed long ideal dsRNA segments onto the hybrids. Assuming that the intervening hybrid segment (dashed line, not modelled) is sufficiently flexible to permit duplex bending to accommodate the curvature, this rudimentary visualization estimated that approximately 60 bp-long hybrid duplex may allow both Fabs from an IgG to bind simultaneously. Since most naturally occurring R-loops are much longer (median R-loop length is 1.5 kb), S9.6 IgGs in cells likely have no trouble engaging both Fabs with R-loop regions.

What is puzzling is that the structure does not fully explain the RNA/DNA sequence preference of S9.6 which had been reported before and which is very apparent in the binding experiments

presented in this work. This mostly likely relies on the dynamic properties of the nucleic acid and could be further explored (beyond the scope of the current work) using molecular dynamics simulations followed by biochemical studies.

A: Indeed, MD simulations will likely bring valuable insights into the distinct helical geometries and structural dynamics of hybrids of different sequences. We are certainly interested in pursuing these in a follow-up study.

In summary, this is a solid and important study and this reviewer does not find any weaknesses of this work.

Minor:

Line 69: Should be “senataxin”

Fixed.

Line 377: Should be “...provides a framework...”

Fixed.

Reviewers' Comments:

Reviewer #1:

Remarks to the Author:

The authors have addressed all my concerns adequately. The only minor comment is that Fig 3a is confusing when they eliminate a residue after rotating. It is difficult to follow the rotation when a major landmark is gone.

Reviewer #2:

Remarks to the Author:

The authors have a good job incorporating my and other reviewer's suggestions. In addition, the manuscript now paints a more clear picture regarding where the Fab largely discriminates between hybrid and dsDNA based on the difference in duplex conformation with individual contacts being less crucial for this selectivity. I think the mutations we and others suggested helped in that regard and resulted in a valuable finding.

I did notice a mislabeled figure: On Fig. 3e the "Hybrid with a single 3 consecutive 2'-OH" and "Hybrid with alternating riboses and deoxyriboses" are swapped with respect to how they're described in the text (lines 226-231).

Reviewer #1 (Remarks to the Author):

The authors have addressed all my concerns adequately. The only minor comment is that Fig 3a is confusing when they eliminate a residue after rotating. It is difficult to follow the rotation when a major landmark is gone.

Authors (A): We thank the reviewer for their positive assessment and detailed guidance in clarifying and improving our manuscript. As suggested, we have now added Y101 back to show this residue as a landmark in both views of Fig 3a.

Reviewer #2 (Remarks to the Author):

The authors have a good job incorporating my and other reviewer's suggestions. In addition, the manuscript now paints a more clear picture regarding where the Fab largely discriminates between hybrid and dsDNA based on the difference in duplex conformation with individual contacts being less crucial for this selectivity. I think the mutations we and others suggested helped in that regard and resulted in a valuable finding.

A: We thank the reviewer for their favorable assessment and helpful suggestions which led to significant improvement of the manuscript and new findings.

I did notice a mislabeled figure: On Fig. 3e the "Hybrid with a single 3 consecutive 2'-OH" and "Hybrid with alternating riboses and deoxyriboses" are swapped with respect to how they're described in the text (lines 226-231).

We thank the reviewer for catching this error and have fixed it. Fig 3e was correctly labeled but the in-text descriptions were inadvertently swapped — now fixed.